# Joint modeling of cellular heterogeneity and condition effects with scPCA in single-cell RNA-seq
Harald Vöhringer [1,2,3,4] ✉

Single-cell RNA sequencing (scRNA-seq) in multi-condition experiments enables the systematic assessment of treatment effects. Analyzing scRNA-seq data relies on linear dimensionality reduction (DR) methods like principal component analysis (PCA). These methods decompose high-dimensional gene expression profiles into interpretable factor representations and prototypical expression patterns (components). However, integrating study covariates within linear DR frameworks remains a challenging task. We present scPCA, a flexible DR framework that jointly models cellular heterogeneity and conditioning variables, allowing it to recover an integrated factor representation and reveal transcriptional changes across conditions and components of the decomposition. We show that scPCA extracts an interpretable latent representation by analyzing unstimulated and IFNß-treated PBMCs and show its utility in mitigating batch effects. We examine age-related changes in rodent lung cell populations, uncovering a previously unreported surge in *Ccl5* expression in T cells. We illustrate how scPCA may be employed to identify coordinated transcriptional changes across multiple time-points in depolarized visual cortex neurons. Finally, we show that scPCA elucidates transcriptional shifts in CRISPR-Cas9 chordin knockout zebrafish single-cell data despite large difference cell abundance across conditions. scPCA is a general method applicable beyond scRNA-seq to other high-dimensional datasets.

Multi-condition single-cell RNA sequencing (scRNA-seq) provides a comprehensive way to analyze how treatments impact various cell types systematically. However, integrating and interpreting single-cell data is challenging, especially if the biological question requires assessing cells across multiple conditions and technical replicates. This complexity stems from the inherently high-dimensional nature of the data and the influence of two primary sources of variation: cellular heterogeneity and experimental perturbation[1].

Central to the analysis workflow for scRNA-seq data are linear dimensionality reduction (DR) methods. These methods decompose high-dimensional gene expression matrices into two smaller matrices: the factor matrix, which provides a low-dimensional representation of cells used for clustering[2] and data visualization techniques[3,4], and the loading matrix, which contains basis vectors or components capturing prototypical expression patterns. However, DR methods like PCA struggle to disentangle variation from cellular heterogeneity and experimental intervention[5,6], often requiring post-hoc adjustments to the factors[6,7].

While some linear DR methods model conditioning variables, they were not intended to capture fully integrated factors, meaning they fail to represent the same cell types from different conditions with similar latent representations. Additionally, these methods do not generalize to experiments with multiple conditioning variables[8] (e.g., genetic perturbation and drug stimulation).

For example, contrastive PCA[9,10] identifies components by contrasting the variance in a target dataset against a background dataset, thus highlighting unique patterns specific to the target. Multi-omics factor analysis (MOFA+)[11] uses sparsity priors to identify subsets of factors that explain condition-specific observations. Both approaches identify condition-specific factors but do not examine how components vary across conditions. The Gene Expression Decomposition and Integration (GEDI)[8] framework enables the extraction of an integrated factor representation by accounting for component changes across conditions. However, GEDI's fixed parameterization, which may be considered as a specific case of the more general method presented here, may only account for a single

[1]European Molecular Biology Laboratory (EMBL), Heidelberg, Germany. [2]Molecular Medicine Partnership Unit (MMPU), MMPU Heidelberg, Heidelberg, Germany. [3]Department of Medicine V, Hematology, Oncology and Rheumatology, University Hospital Heidelberg, Heidelberg, Germany. [4]Department of Hematology and Oncology, University Hospital Düsseldorf, Düsseldorf, Germany. ✉e-mail: harald.voehringer@embl.de

conditioning variable at a time (for a more comprehensive overview, see also the Supplementary Methods).

The problem of disentangling latent spaces with respect to conditioning variables has been explored in the context of non-linear low-rank representation learning[12]. For example, conditional variational autoencoders[13] (cVAEs) retrieve integrated data representations by informing the model about the conditioning variable[14–16]. However, unlike linear DR methods, where each component can be interpreted as a "metagene", the encoder and decoder functions in cVAEs are less straightforward to interpret.

We developed scPCA, a flexible factorization model for analyzing multi-condition single-cell datasets. Our model incorporates conditioning variables, enabling scPCA to extract condition-specific bases. This allows analysts to assess expression shifts across each condition and component and retrieve an integrated factor representation of the data. We exemplify how scPCA extracts a low-rank representation that captures the main drivers of variation by applying our tool to untreated and IFNß-stimulated PBMCs[17]. We showcase how our factorization model may be employed to address batch effects by integrating cell-line data, and assess it in various single-cell batch integration benchmarking tasks[18]. We illustrate that scPCA factors often discriminate cell-type clusters by analyzing lung cells of aging mice, and demonstrate that their corresponding components may be leveraged to study their constituent gene programs. We then apply scPCA to a dataset comprising multiple timepoints to reveal early and late transcriptional responses in various cell populations of depolarized rodent brain cells. Finally, we show that scPCA faithfully recovers the effects of a CRISPR-Cas9 chordin knockout in zebrafish embryos despite uneven cell type distributions across conditions.

## Results

### Disentangling experimental states and cellular heterogeneity

In scPCA, we consider single-cell expression count matrices with categorical covariates indicating conditions such as drug treatment[17] or genetic perturbation[19,20] (Fig. 1a, b). These conditioning variables are represented as a full-rank design matrix, used to parameterize a loading weight tensor, capturing the condition-specific components relative to the basis of a user-defined reference condition (Fig. 1c, "Methods"). Additionally, we model a mean offset matrix, which can optionally be parameterized using an indicator matrix, allowing scPCA to account for condition-specific shifts in the data space. This adjustment is sometimes necessary because different conditions can cause global shifts in gene expression profiles, leading to distinct data distributions that must be properly aligned to accurately extract the principal components governing the data (Fig. 1c, "Methods").

Multi-condition single-cell datasets are characterized by shared cell types that exhibit altered expression profiles because of experimental interventions. For this reason, we hypothesized that such data are governed by closely related but not identical lower-dimensional principal axes of variation. To illustrate this concept, we generated multi-condition data in $\mathbb{R}^2$ using two distinct bases to model 1-dimensional condition-specific data subspaces, and simulated two clusters to represent two shared cell types (Fig. 1d, "Methods"). Note that cell cluster A from the treatment (trt) condition has a higher propensity to express gene $x_2$ in comparison to its counterpart from the reference group (Fig. 1d). This demonstrates how systematic shifts in gene expression across one or multiple genes can introduce additional variability within cell type clusters, even when the same cell types are present across different conditions (Fig. 1d).

Modeling single-cell data without explicitly accounting for its experimental structure leads to the extraction of a basis that fails to reconstruct the true data-generating subspaces (Fig. 1e). This results in large residuals between observed and reconstructed data points (Fig. 1e, gray lines). Consequently, DR methods explain the variation induced by the condition with additional factors leading to a lower-dimensional representation of the data that conflates variation due to cellular heterogeneity with variability attributed to the experimental intervention. Notably, the existence of such factors impedes subsequent downstream analyses, given that the factor

representation serves as the foundation for downstream tasks such as clustering[21,22] and visualization[3,4].

Conversely, in scPCA, we factor out the variability attributed to the experimental intervention by allowing the model to extract distinct, yet aligned condition-specific basis vectors (hereafter scPCs). This results in a more accurate representation of the data-generating subspaces, and thus to smaller residuals between observed data points and corresponding projections (Fig. 1f). As a result, scPCA factors explain a greater proportion of the variance compared to other factor models at equivalent decomposition ranks and eliminate the necessity for factors that explicitly address variance attributed to the conditioning variable (Fig. 1f). Finally, due to the linearity of the model, a scPCA decomposition enables the analyst to compute loading weight differences (LWDs), which highlight changes in individual scPCs across conditions, providing insights into how gene expression varies in each component.

We implemented scPCA as a hierarchical Bayesian factor model and employed stochastic variational inference to guarantee scalability to large single-cell datasets[23–25]. To ensure robust inference, we employ an over-dispersed Negative Binomial distribution to model the count expression matrix (see "Methods" section)[26,27], and tested the model extensively in simulation experiments (Supplementary Fig. 1a–c). We further benchmarked scPCA against other single-cell latent variable models using both simulation experiments (Supplementary Fig. 7a–c) and a diverse set of published datasets (Extended Data Figs. 7 and 8). Across these analyses, scPCA consistently demonstrated competitive performance across a comprehensive range of evaluation metrics. Notably, scPCA scales efficiently with increasing dataset size, exhibiting competitive runtime and memory usage compared to other methods[5,6,15] (Supplementary Fig. 7d). Taken together, scPCA's flexible factorization framework effectively deconvolves the principal axes of variation with respect to experimental covariates.

### Accounting for interferon-beta stimulation in PBMCs

To assess the efficacy of scPCA in integrating and interpreting multi-condition single-cell datasets, we employed our tool on public scRNA-seq data of PBMCs obtained from 8 lupus patients. These cells were either activated with recombinant IFNß (stim) or left untreated (ctrl) for a duration of 6 h. The dataset comprises a total of 24,305 cells, roughly divided into equal parts of treated (12,138) and untreated (12,167) single cells[17].

The strong perturbation effect of IFNß induces widespread transcriptomic alterations in all cells[5]. As a result, applying the standard scRNA-seq analysis pipeline, which included PCA and uniform manifold approximation and projection[3] (UMAP), led to cell clustering based on both cell type and condition, complicating data interpretation (Fig. 2a). Upon analyzing PCA factors, we found that the first factor determined the lymphoid–myeloid axis, while the second factor separated the cells by condition, explaining the poor integration of the data (Fig. 2d).

Subsequently, we applied a condition-informed scPCA decomposition to the data, allowing the model to fit condition-specific components (Supplementary Fig. 2a, "Methods" section). Notably, the UMAP based on scPCA factors indicated a successful integration of the dataset (Fig. 2b, c), which was further supported by integration and biological conservation scores (Supplementary Fig. 2h)[6,18]. Akin to PCA, we observed that the scPCA Factor 1 separated cells based on lineage (Fig. 2e). However, scPCA Factor 2 explained variance within lymphatic cells rather than accounting for variation associated with the condition (Fig. 2e), indicating that the model successfully accounted for the variation induced by the drug stimulation.

Next, we investigated whether our algorithm produced a latent representation comparable to PCA when information about the condition is omitted (Supplementary Fig. 2b, "Methods" section). Similarly to PCA, we found that scPCA's cell embeddings generated a UMAP representation wherein ctrl and stim cells separated by condition (Supplementary Fig. 2c). Furthermore, we verified that the first two factors explained lineage and IFNß-stimulation (Supplementary Fig. 2d), establishing these variables as the main drivers of variation in the dataset.

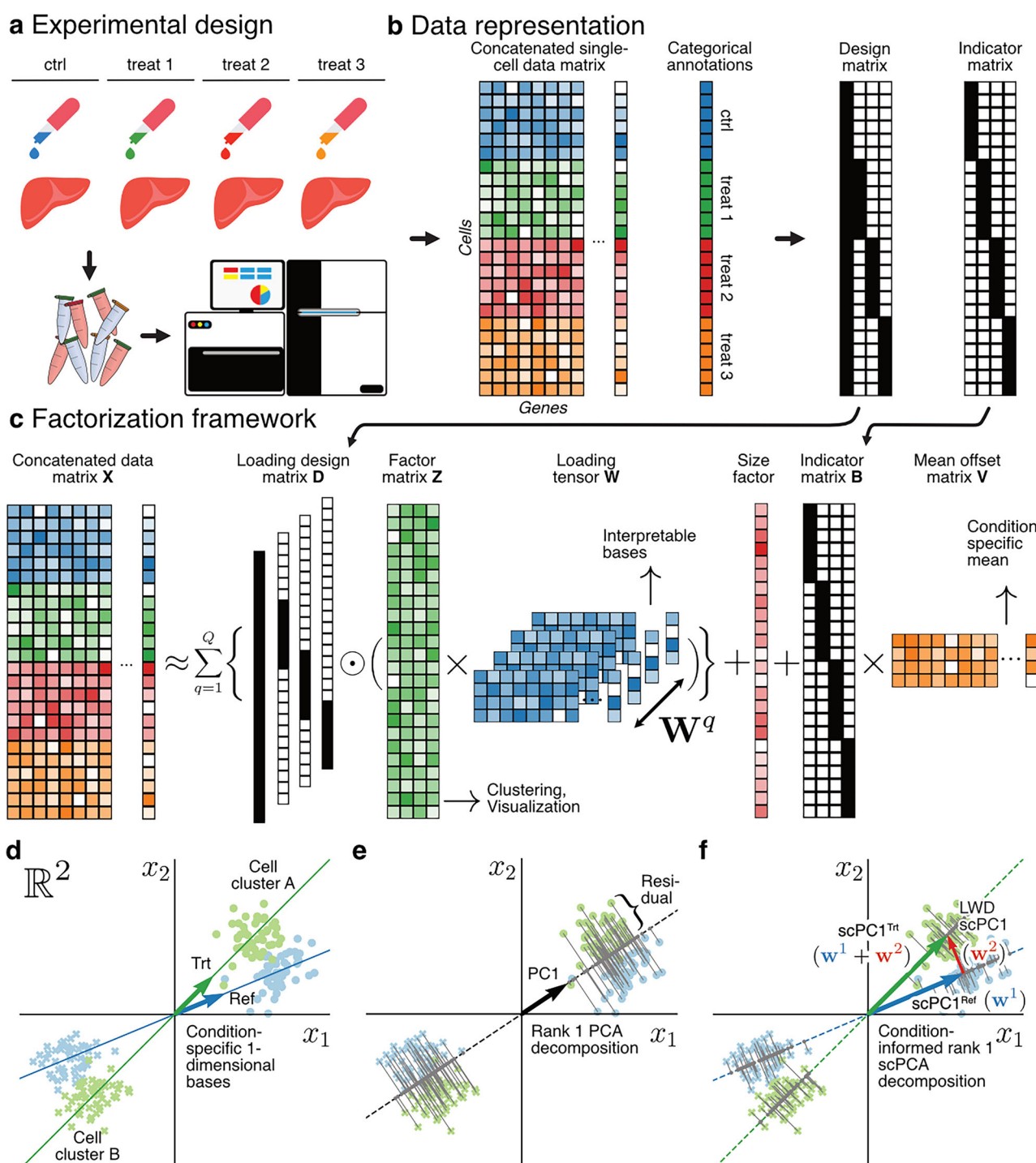

**Fig. 1 | scPCA's factorization framework allows to account transcriptional changes in expression across experimental conditions. a** Schematic depiction of an experimental design involving multiple treatments. **b** A single-cell expression count matrix may be obtained by concatenating the expression matrices of each experimental condition. Categorical covariates may be leveraged in scPCA in the form of design and indicator matrices. **c** Schematic depiction of the factor model (the $\odot$ symbol denotes element-wise multiplication). The loading design matrix parameterizes a loading tensor, enabling the model to identify condition-specific bases. An optional indicator matrix may be employed to fit different mean offsets to each experimental condition. **d** Schematic depiction of simulated single-cell data in 2-dimensional (log) gene space. Arrows and lines indicate the data-generating 1-dimensional bases and subspaces of the reference (blue) and treatment (green)

conditions, respectively. **e** Ignoring the conditioning variable leads to the extraction of a basis that fails to reconstruct the principal subspaces of the reference or treated condition. Consequently, conventional factor models such as PCA explain the residual variance with additional factors (not shown). **f** Incorporating the loading design matrix in the scPCAs factorization framework enables the extraction of condition-specific basis vectors (scPCs). In **c–e** dots and crosses indicate two cell clusters. Green and blue data points highlight the reference and treatment conditions. Image credits: sequencing machine and Eppendorf tube from Openclipart (https://openclipart.org/detail/317807/sequencing-machine, https://openclipart.org/detail/218545/opened-eppendorf-tube-red), liver image and pipette logo adapted from SVG Repo (https://staging.svgrepo.com/svg/120859/liver, https://www.svgrepo.com/svg/275988/pipette).

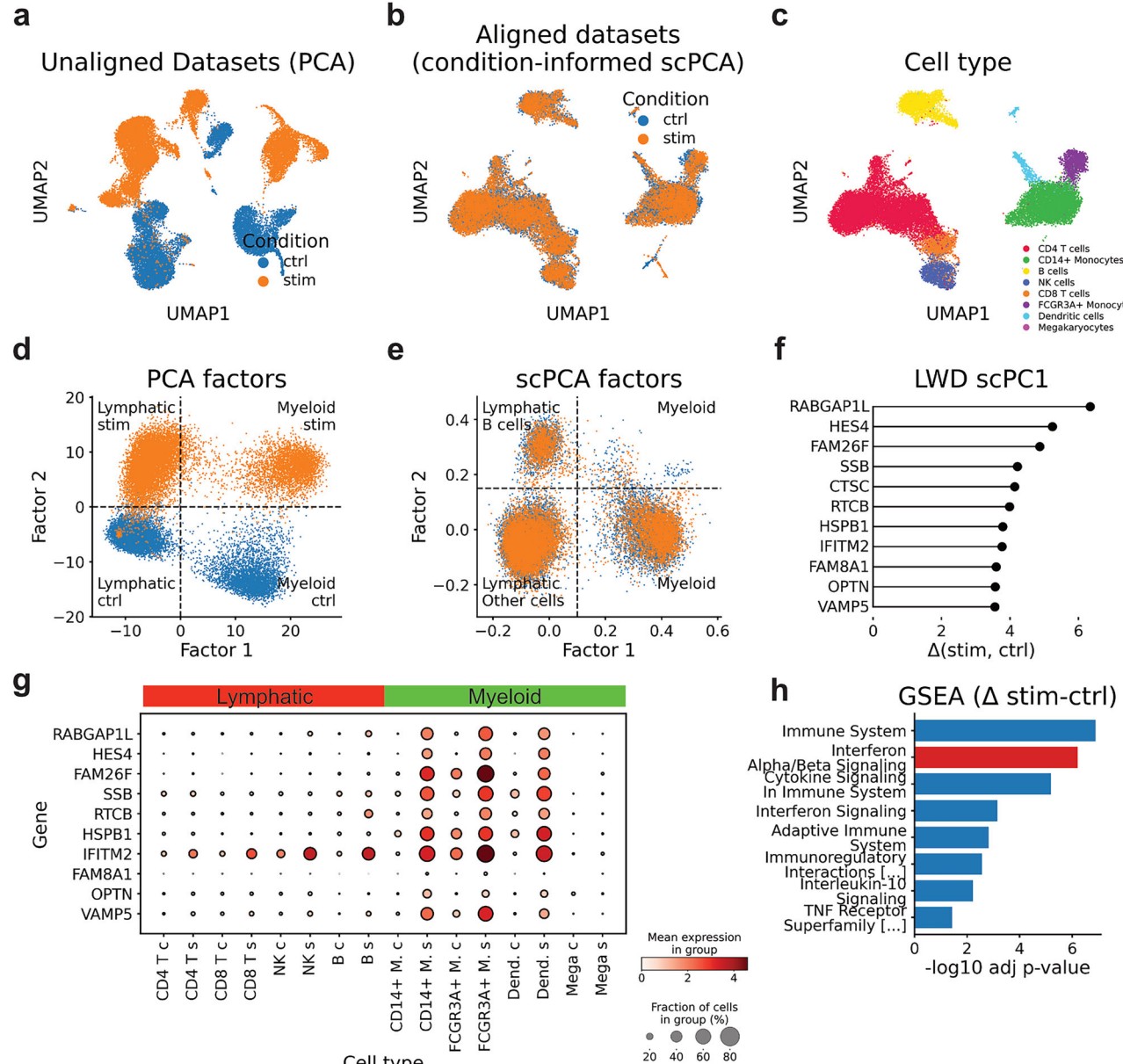

**Fig. 2 | Modeling control and IFNß stimulated PBMCs with scPCA. a** UMAP visualization showing untreated (control) and IFNß stimulated (stim) PBMCs ($n = 24{,}305$), derived from a rank 20 PCA decomposition. **b, c** UMAP plots generated from a condition-informed rank 20 scPCA decomposition, with annotations for condition (**b**) and cell types (**c**). **d, e** Scatter plots illustrating the first two factors from PCA (**d**) and scPCA (**e**). **f** Positive scPC1 LWDs indicate genes with the propensity to be more highly expressed in stimulated myeloid cells. **g** Dot plot showing genes with the largest scPC1 LWDs (**f**) across cell types and conditions (the abbreviations c and s denote control and stimulated cells, respectively). **h** A gene set enrichment analysis (GSEA) based on the top LWDs of scPC1 indicates upregulated gene programs in IFNß-stimulated myeloid cells, including IFNß signaling (highlighted).

We then analyzed scPCA's loading tensor to better understand how the model accounted for the two conditions. The cross-correlation of components revealed minimal similarity among the scPCs, implying that the basis vectors were largely orthogonal to each other (Supplementary Fig. 2e). However, a closer inspection of the same component across conditions (diagonal terms) revealed high correlations, suggesting that the scPCs were largely preserved across conditions (Supplementary Fig. 2f). These observations indicate that while different scPCs capture largely orthogonal sources of variation, the same scPC captures similar sources of variation across conditions.

We proceeded to analyze the first components of the scPCA factorization. Subjecting the top gene weights of scPC1 to gene set enrichment analysis (GSEA) revealed associations with gene sets related to the innate immune system (Supplementary Fig. 2g), consistent with Factor 1's

association with myeloid cells (Fig. 2e). To understand how IFNß altered the gene expression profile of myeloid cells, we computed scPC1 LWDs across conditions (Fig. 2f, "Methods" section). This analysis revealed the expected upregulation of genes involved in cytokine-mediated pathways, including *IFITM2*, *ISG20*, and *CXCL10*. However, the largest LWDs were observed in genes such as *RABGAP1L*, *HES4*, and *FAM26F*, which are associated with protein translocation (Fig. 2f). This suggests altered protein metabolism in myeloid cells because of the stimulation[28]. The analysis of gene expression confirmed that these genes were upregulated in stimulated myeloid cells, but not in other cell types (Fig. 2g). Finally, a GSEA based on scPC1 LWDs revealed an activated IFNß process, providing evidence that LWDs accurately capture activated gene programs (Fig. 2h). Taken together, these observations suggest that LWDs capture cell-type-specific changes in gene expression across conditions.

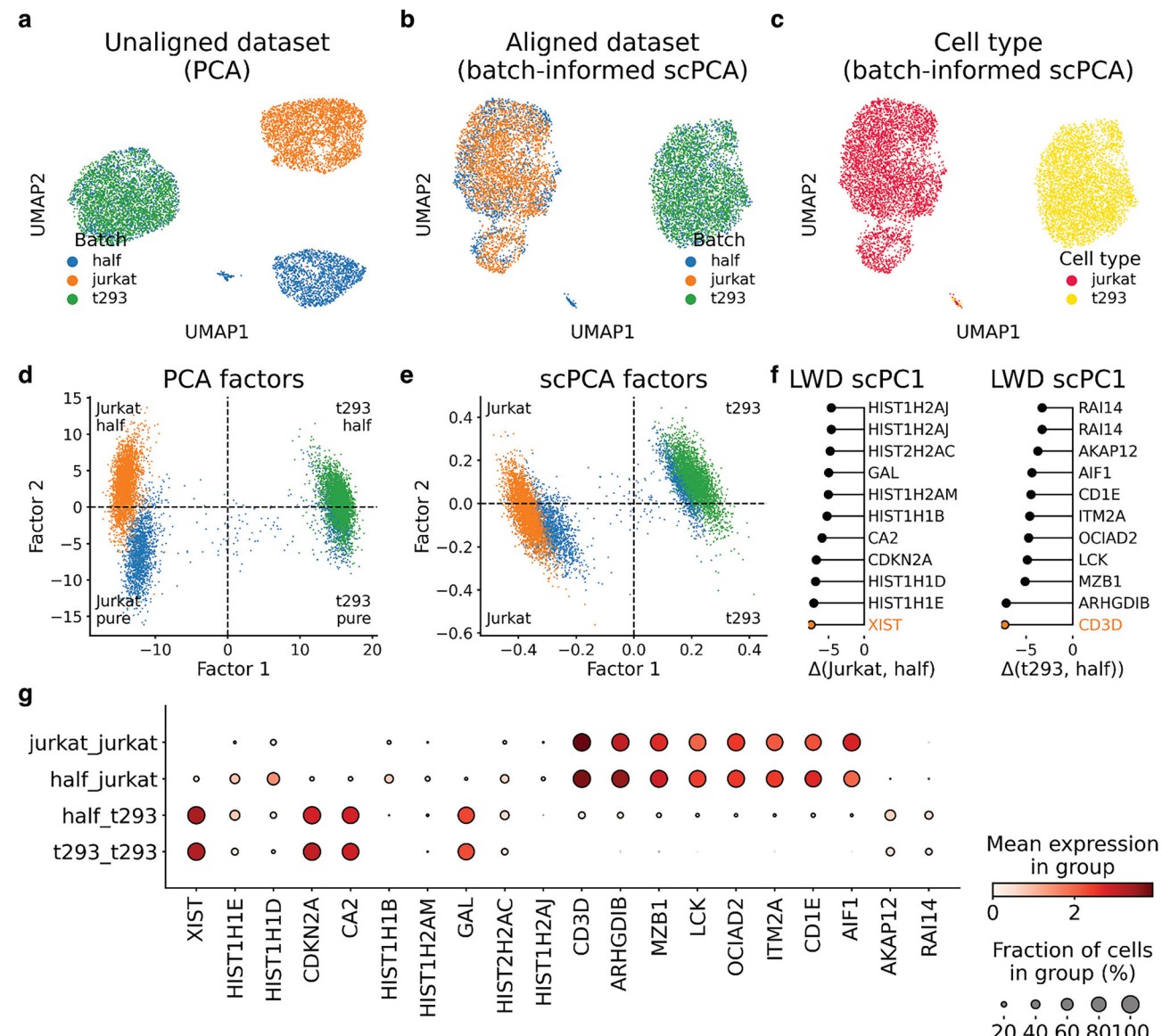

**Fig. 3 | Modeling batch effects in cell-line data with scPCA. a** UMAP plot showing pure Jurkat and t293 cells, and a mixture of Jurkat and t293 cells (half) on the basis of a rank 20 PCA decomposition. **b, c** UMAP plots based on a rank 20 batch-informed scPCA decomposition with annotations for the batch (**b**) and cell line (**c**). **d, e** Scatter plots of factor 1 and 2 from the PCA (**d**) and the batch-informed scPCA (**e**) factorization. **f** Negative scPC1 loadings indicate genes that tend to be expressed at lower levels in the pure cell-line batches compared to the half mixture batch. **g** Dot plot displaying the genes with the largest negative scPC1 LWDs across cell lines and batches.

**The flexible scPCA framework enables modeling batch effects**

Single-cell RNA-seq studies often comprise multiple experiments with differences in capturing times, experimenters, reagents, equipment, and technology platforms[29]. These variations can introduce batch effects, reflecting variability due to technical artifacts rather than biological differences. There is a large body of literature on computational methods addressing the challenge of integrating such datasets, which rely on approaches ranging from mutual nearest neighbors[1], matrix factorization[6,30], or more complex models based on neural networks[14,31]. However, none of the aforementioned methods offer an interpretable explanation for batch effects, leading to concerns about potential overcorrection[18,29].

We hypothesized that scPCA's ability to incorporate conditioning variables could help to account for batch variation in single-cell data. For this reason, we used previously published single-cell data[32], comprising a 50/50 mixture ($n = 3364$) of Jurkat and t293 cells, and pure batches of Jurkat ($n = 5054$) and t293 ($n = 2859$) cells, and attempted to model them jointly using scPCA. Notably, Jurkat cells, which are derived from a male cell line, predominantly express the *CD3D* gene, whereas t293 cells, originating from a female cell line, exhibit a pronounced expression of the X-chromosome silencing gene *XIST*[82].

As previously noted[6], using PCA embeddings as input for UMAP resulted in unsatisfactory outcomes, wherein only t293 cells from the mixture and pure batch integrate (Fig. 3a). A closer examination of PCA factors indicated that Factor 1 was likely to distinguish both cell lines, while Factor 2 assigned positive and negative weights to Jurkat cells from the pure and half batch, respectively (Fig. 3d).

Next, we applied scPCA to the dataset by selecting the half-batch as reference condition (Supplementary Fig. 3b). Our findings indicated that the scPCA factorization effectively addressed the batch effect, as Jurkat and t293 cells from the mixed and the pure batches mixed in the UMAP representation of the data (Fig. 3b, c) and exhibited better integration metrics in comparison to PCA (Supplementary Fig. 3e). Further, we confirmed the lack of a factor with a clear tendency to explain the batch

structure (Fig. 3e), thus providing an explanation for the successful data integration.

We then ran scPCA in a batch-agnostic manner to assert that the model extracts a latent representation akin to PCA when information about the batch structure is omitted (Supplementary Fig. 3b, "Methods" section). We found that the UMAP constructed from the uninformed scPCA model failed to account for the batch effect (Supplementary Fig. 3c). Similar to PCA, we found that Factor 2 captured batch differences (Supplementary Fig. 3d).

To elucidate the genes responsible for the poor integration of Jurkat cells, we focused on scPC1, which delineated Jurkat from t293 cells (Fig. 3e). Analyzing the scPC1 LWDs between the pure Jurkat and the half-half mixture batch exposed genes with notably negative magnitude, suggesting high expression in cells of the half batch but not in pure Jurkat batch cells (Fig. 3f). Among these genes, we identified the t293 hallmark gene *XIST*, as well as *CDKN2A* and *CA2* which are highly expressed in t293 cells but not in Jurkat cells (Fig. 3g). Interestingly, scPCA detected histone encoding genes (*HIST1H1D*, *HIST1H1E* etc.) which were upregulated in Jurkat cells of the half batch but not in the pure Jurkat batch (Fig. 3f). Conversely, inspecting the LWDs between the pure t293 and 50/50 mixture batch of scPC1 (Fig. 3f) indicated the down-regulation of Jurkat-specific genes, including its hallmark gene *CD3D* (Fig. 3f, g).

To assess scPCA's ability to model batch effects in more complex scenarios, we applied scPCA to a variety of benchmarking and published datasets (Extended Data Figs. 1–8)[18]. We observed that scPCA integrated data across different platforms (Extended Data Fig. 1b, d). Applying scPCA on a single-cell dataset of immune cells collected from multiple donors yielded an integrated data representation as visually confirmed through UMAP inspection (Extended Data Fig. 2a), but interestingly resulted in lower batch integration scores in comparison to baseline PCA (Extended Data Fig. 2d). More complex datasets, such as Luecken et al.'s lung and immune mouse dataset, which contained samples from different spatial locations, laboratories, protocols, and species, presented integration challenges (Extended Data Figs. 3 and 4). In these tasks, scPCA's linear framework was not sufficient to account for technical variation, a limitation that was visually evident through UMAP inspection. Finally, we tested scPCA on the two simulated single-cell datasets, which were characterized by variations in cellular composition and nested batch effects, respectively (Extended Data Figs. 5 and 6). Here, we found that scPCA performed well on the simulated data set containing variable cellular composition but encountered difficulties in integrating the dataset with nested batch effects.

In conclusion, our evaluation of scPCA across various benchmarking datasets demonstrated its robustness in handling batch effects tasks such as integrating data across platforms and handling datasets with multiple donors. However, scPCA's linear model exhibited limitations when addressing more complex batch covariates, such as spatial locations, laboratories, protocols, and species. These findings highlight the limitations of linear scRNA-seq integration methods in more intricate scenarios.

## scPCA extracts cell-type-specific factors in complex sc-datasets

To assess scPCA's utility in more complex single-cell data, we applied scPCA to lung samples of aging mice[33], which included more than 20 different cell types. This dataset contained 7639 and 6425 cells (total $n = 14,064$) from mice sacrificed at 3 (3 m) and 24 months (24 m), respectively.

To account for the age of mice, we chose the 3 m condition as the reference condition and modeled changes in cells of 24 m old mice as deviations thereof (Supplementary Fig. 4a). Visual inspection of the scPCA-based UMAP indicated the integration of cells across conditions (Fig. 4a, b). We proceeded to analyze the factors, which revealed that Factors 2, 3, 5, and 6 displayed either high positive or negative weights in pneumocytes, ciliated cells, alveolar macrophages, and T cells, respectively (Fig. 4c).

Next, we subjected the top-loading weights of scPC2, 3, 5, and 6 to GSEA, which revealed an enrichment of cell-type-specific gene sets for each

factor. For example, scPC2 was linked to two processes related with surfactant metabolism, aligning with its association with type II pneumocytes (Supplementary Fig. 4b). Likewise, scPC3 captured processes such as intraflagellar transport and cilium assembly, consistent with its enrichment with ciliated cells (Supplementary Fig. 4c). As anticipated, Factor 5 and 6 revealed gene sets associated with the immune system (Supplementary Fig. 4d), with the latter including pathways explicitly linked to T cells (Supplementary Fig. 4e).

To evaluate the impact of aging, we next computed LWDs across scPCs (Fig. 4d–g). Among the top upregulated genes of scPC2 were the MHC I-associated gene *H2-Q7* and the Acyl-CoAl desaturase *Scd1* (Fig. 4d, h). Conversely, downregulated genes of scPC5 included *Ear1* and *Ear2* (Fig. 4e, h), corroborating the original authors' findings for type II pneumocytes and alveolar macrophages[33]. The examination of scPC3 LWDs, representative for ciliated cells, indicated an upregulation of genes associated with cellular stress response, including *Nupr1* and *Cyp5a5* (Fig. 4f, h)[34,35]. Finally, changes along scPC5 indicated genes integral to immune response and regulation (Fig. 4g, h). These included *S100a6*, known for its role in cell stress[36,37]; *Ccl5*, a chemokine gene involved in inflammatory responses[38]; and *Il8r1*, which encodes a receptor important for immune cell migration[39].

We next applied GSEA to genes associated with the largest LWDs. This analysis revealed processes predominantly associated with oxidative stress and immune response (Supplementary Fig. 4f–i), aligning with the concept of ongoing 'inflammaging' in aging organisms[40]. For example, Factor 3, linked to ciliated cells, exhibited an enrichment of processes linked to oxidative stress (Supplementary Fig. 4g).

In summary, we find scPCA's factor representation and its capability to model experimental states valuable for exploratory data analysis. In the context of single-cell data analysis, scPCA factors frequently align with cell-types, encapsulating their gene-expression programs within the associated components. Moreover, the linear parameterization of the loading tensor facilitates the interpretation of how individual scPCs change across conditions, thereby offering insights into either up- or down-regulated gene programs, even in scenarios where the perturbation results in nuanced effect sizes.

## Modeling multiple time points in scPCA

We then explored how scPCA disentangles single-cell datasets with conditioning variables including more than two levels. Particularly, we examined single-cell data from the visual cortex of mice that had been kept in darkness for seven days before being subjected to light exposure for durations of 0 (control), 1, and 4 h (Fig. 5a)[41]. The dataset comprised a total of 48,266 single cells, distributed across the conditions with 16,188; 13,571; and 18,507 cells, respectively.

Exposure to depolarizing stimuli is thought to initiate two distinct phases of transcriptional changes. The first phase is controlled by early-response transcription factors (ERTFs), which are responsible for initiating the expression of late-response genes (LRGs)[42,43]. Interestingly, various types of neuronal cells follow a similar ERTF program, whereas the expression of LRGs, encompassing neuronal modulators and secreted factors, varies depending on the cell type.

We applied scPCA to the dataset by modeling the time points 1 h and 4 h with respect to the 0 h reference condition (Supplementary Fig. 5a). Subjecting the scPCA factors to UMAP DR indicated the integration across time points (Fig. 5a, b). A subsequent analysis of factors revealed that Factors 1, 2, 3, and 4 showed either high or low values in excitatory and inter-neurons, astrocytes, microglia, and endothelial and smooth muscle cells, respectively (Fig. 5c).

We then applied GSEA to the top-loading weights of scPC1-4. This analysis revealed terms such as neuronal system or transmission across chemical synapses for scPC1, indicative for neuronal cells (Supplementary Fig. 5b). Likewise, scPC3 was characteristic for gene sets related to the immune system, consistent with the association of Factor 3 in microglia (Supplementary Fig. 5d). The GSEAs for scPC2 and 4 revealed gene programs related to metabolism and compound transport, consistent with

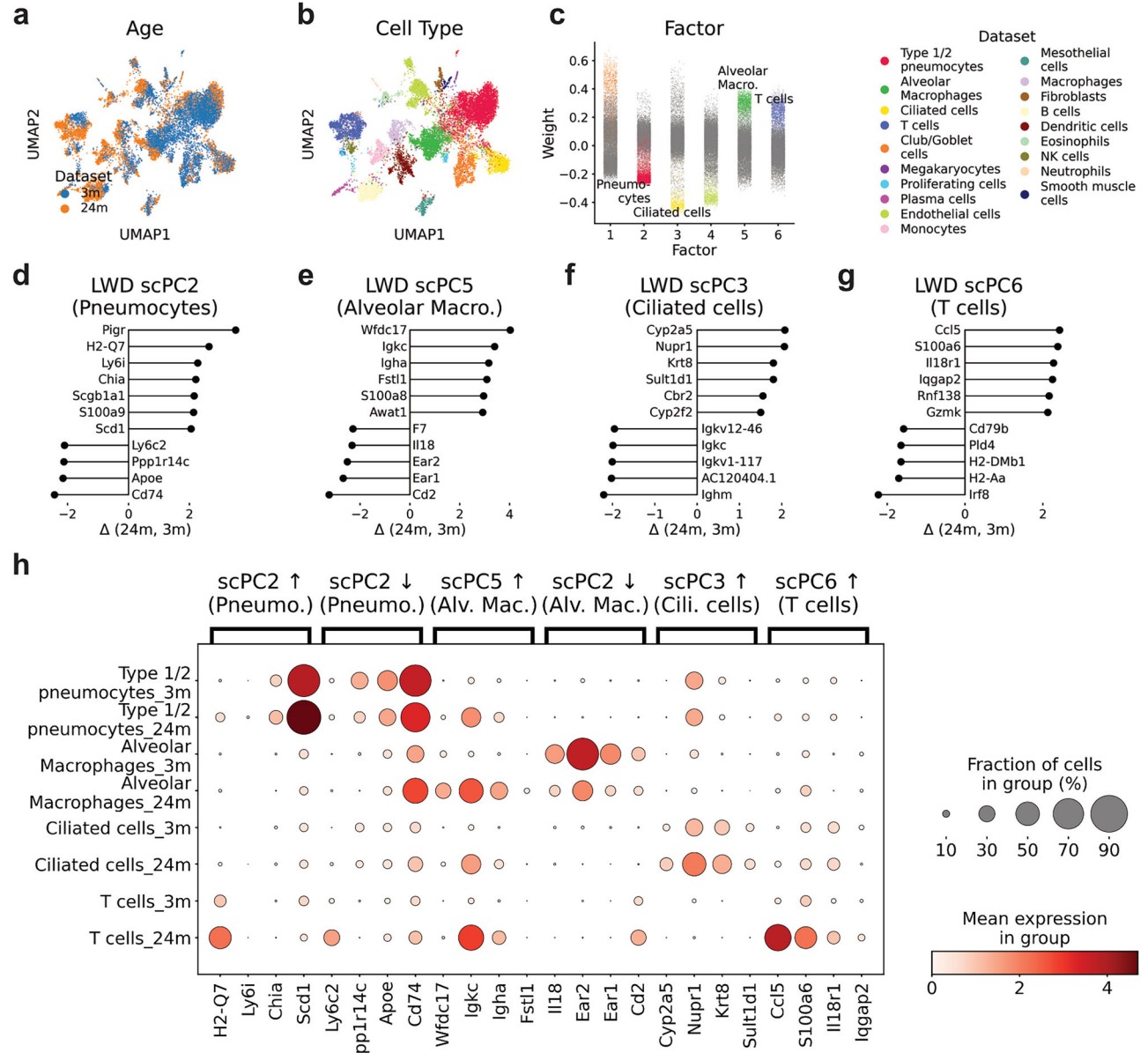

**Fig. 4 | The scPCA decomposition sheds light on transcriptional changes in lung cells of aging mice. a, b** UMAP plots based on scPCA cell embeddings with annotations for the condition (**a**) and cell type (**b**). **c** The scPCA factors show high factor values for certain cell types. **d–g** LWDs of scPC2, 5, 3, and 6 indicate genes that are either up- or down-regulated in aging mice. **h** Dot plot displaying selected genes with the largest LWDs (**e, f**).

roles of astrocytes in metabolic support to neurons and endothelial cells in barrier and transport functions within the brain (Supplementary Fig. 5c, e).

Next, we computed LWDs for each scPC across the time points 0 h and 1 h. As anticipated, the examination of the top 10 LWDs of scPC1, enriched in excitatory and interneurons, revealed an upregulation of neuronal-specific ERTFs such as *Egr2*, *Egr4*, and *Fosb*, as well as common ERTFs like *Nr4a2* and *Fosl2* (Fig. 5d)[41]. Highest ranking LWDs of scPC2, associated with astrocytes, included non-neuronal ERTFs such as *Klf4*, alongside common ERTFs like *Nr4a1*, *Nr4a2*, *Nr4a3*, *Fos*, and *Fosl2* (Fig. 5e)[41]. Conversely, we detected only moderate loading weight changes and no ERTFs among scPC3 LWDs, consistent with the observations made by the original authors for microglia (Fig. 5f). Finally, scPC4 LWDs displayed non-neural ERTFs like *Klf4* and *Maff*, and shared ERTFs including *Nr4a1*, *Nr4a2*, *Nr4a3*, *Fos*, and *Fosl2* (Fig. 5g).

We then subjected the top LWDs of scPC1-4 across the 1 h and 0 h conditions to GSEA. Consistent with a shared ERTF response across cell types, this analysis revealed similar gene programs for scPC1, 2, and 4 LWDs, with the process "Signaling by NTRK1" being common to all of them (Supplementary Fig. 5f, g, i). Indeed, *NTRK1* is a receptor tyrosine kinase associated with the Ras/MAPK pathway, which is activated upon a depolarizing stimulus. Beyond the set of shared terms, we observed that each component's gene sets also encompassed distinctive processes, which may mirror processes triggered by cell-type-specific ERTFs. For example, scPC1 LDWs were indicative of signal transduction, whereas scPC4 LWDs were associated with pathways promoting cell survival and angiogenesis (Supplementary Fig. 5f, i).

We proceeded to investigate the longer ranging response to light stimulation by calculating LWDs between the 0 h and 4 h conditions. Expectedly, the top 10 genes with the largest positive LWDs of scPC1 contained the LRGs *Nptx2*, *Bdnf*, *Inhba*, *Cdkn1a*, *Cbln4*, *Car12*, and *Vgf*[41],

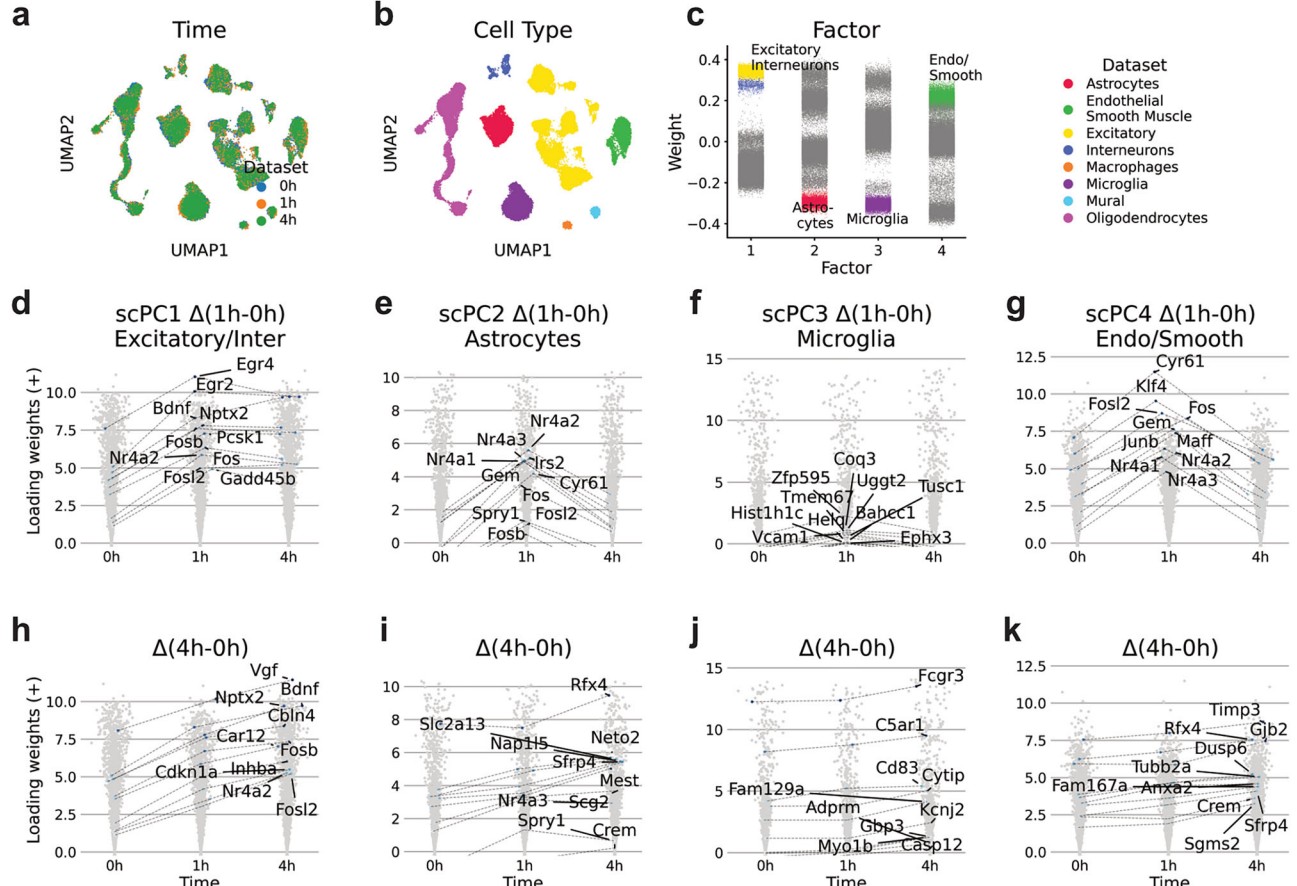

**Fig. 5 | scPCA captures differentially expressed genes in light-stimulated brain cells across multiple time points. a, b** UMAP plots based on scPCA cell embeddings with annotations for the condition (**a**) and cell type (**b**). **c** Factor weights of the first four scPCA components show an association with certain cell types. **d–k** Positive loading weights of scPC1-4 for the 0 h, 1 h, and 4 h time points. In **d–g** and **h–k**, the genes of the ten highest ranking LWDs for the 1 h and 0 h, and the 4 h and 0 h conditions are highlighted, respectively.

indicating that scPCA successfully captured stimulus-induced late transcriptional changes in excitatory and inhibitory neurons (Fig. 5h). Similarly, the top LDWs of scPC2, 3, and 4 captured the LRGs *Rfx4*, *C5ar*, and *Anxa2*, respectively[41] (Fig. 5i-k).

Next, we applied GSEA to the top LWDs of scPC1-4 across the 4 h and 0 h conditions. For scPC1, LWDs indicated ongoing neuronal activity (Supplementary Fig. 5j), which were characterized by gene sets that were also present in the GSEA for the 1 h and 0 h conditions (Supplementary Fig. 5f). The LWDs for scPC2 and scPC4 were indicative of diverse intracellular processes (Supplementary Fig. 5k, m). In contrast to the shared gene sets seen in response to ERTFs (Supplementary Fig. 5f–i), these processes did not exhibit commonalities across cell types, suggesting that the response induced by LRGs is specific to each cell type. Interestingly, scPC3 LWDs displayed the upregulation of various gene sets connected to the innate immune system (Supplementary Fig. 5l). However, as the stimulus-dependent response in microglias was only modest, these results should be verified independently. In summary, we find that the scPCA's factorization framework is capable of taking into account conditioning variables with multiple levels.

**Modeling single-cell perturbation data with scPCA**

In our final exploration, we aimed to assess whether scPCA effectively handles datasets with uneven cell type distributions across conditions (Supplementary Fig. 6b). To address this, we explored zebrafish data wherein the chordin locus was knocked out using CRISPR-Cas9 technology, resulting in zebrafish embryos with expanded ventral tissues[44]. This dataset contained 10,782 control (tyr) and 16,046 chordin (chd) knockout cells (total $n = 26,828$).

We applied our algorithm by modeling chd cells with respect to the reference tyr condition (Supplementary Fig. 6a). Visual inspection of the scPCA-based UMAP indicated the integration of cells across conditions (Fig. 6a, b). We then proceeded to investigate the first three factors, which showed high and low factor values for epidermal, neural, and tailbud cells (Fig. 6c).

We then applied GSEA to the top-loading weights of scPC1–3. For scPC1, this analysis included terms such as "embryonic cranial skeleton morphogenesis" and various other processes linked to the early formation of anatomical structures originating from epidermal cells (Supplementary Fig. 6b). The loading weights for scPC2 and scPC3 included gene sets associated with neuron differentiation and mesoderm development, respectively. This aligns with the enrichment of neural and tailbud cells in these factors (Supplementary Fig. 6c, d).

Next, we assessed the scPC1–3 LWDs to understand the transcriptional changes occurring because of the chd knockout. Interestingly, this analysis indicated an upregulation of *szl* in the top LWDs of scPC1 and 3, but not in scPC2 (Fig. 6d, f). Chordin serves as an inhibitor of bone morphogenic protein (BMP), and *szl* operates as a ventral center molecule that is expressed under conditions of elevated BMP signaling[45]. Consequently, the absence of chordin in knockout cells could lead to an increase in *szl* transcript expression, a change that, to our knowledge, was not noticed in the author's analysis of the data (Supplementary Fig. 6i, j). Additionally, we observed the upregulation of diverse differentiation factors, including *mdka*,

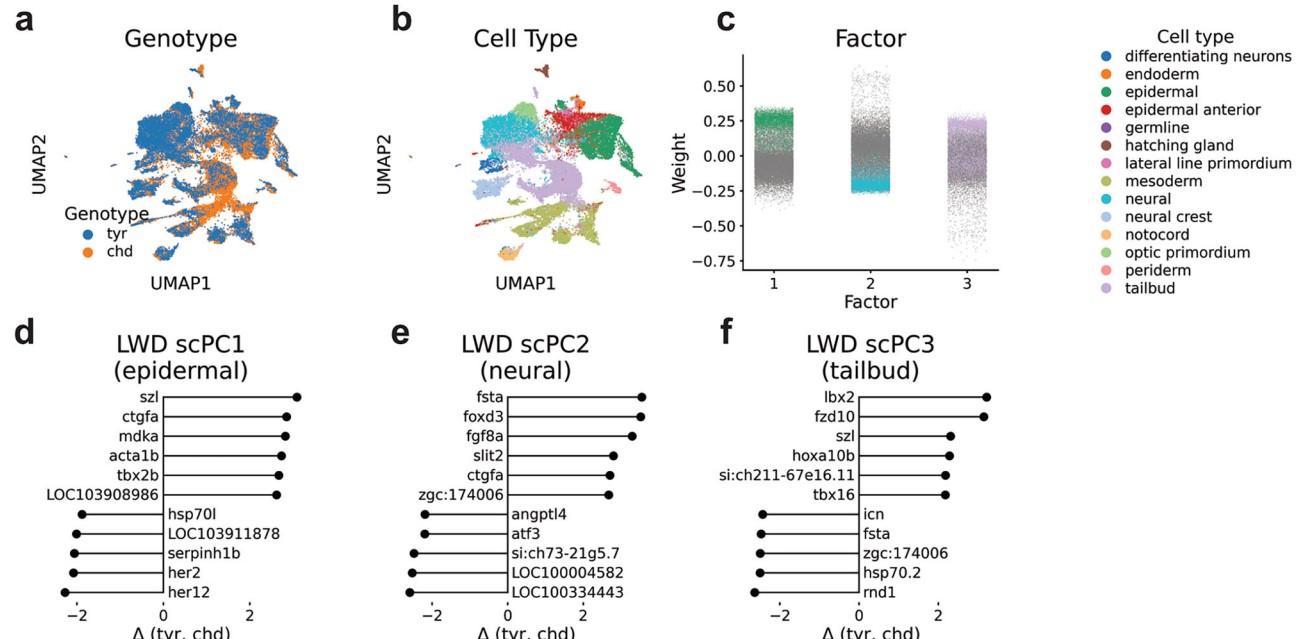

**Fig. 6 | Modeling single-cell perturbation data with scPCA. a, b** UMAP plots based on scPCA cell embeddings with annotations for the perturbation (**a**) and cell type (**b**). **c** Factor weights of scPCA components 1–3. Each dot represents a cell, and colors indicate the cell-type. **d–f** Highest ranking LWDs of scPC1–3 indicate genes that are either up- or down-regulated across conditions.

*tbx2b, fsta, fgf8a, lbx2, fzd10,* and *hox10b* in chd cells, suggesting that the knockout induced widespread changes in the expressed differentiation programme of the developing zebrafish (Fig. 6d–f and Supplementary Fig. 6j). Among the genes exhibiting negative LWDs, we found notch-dependent genes like *her1* and *her12* in scPC1 (Fig. 6d and Supplementary Fig. 6j), indicating that chordin loss affected broader regulatory networks that coordinate embryonic development.

Finally, we subjected scPC1–3 LWDs to GSEA. This analysis revealed gene ontology terms indicative of the development of organs originating from epidermal and neural cells for scPC1 and 2, respectively (Supplementary Fig. 6f, g). In line with the important role of *chordin* in tailbud cells[44], the GSEA for scPC3 LWDs indicated the GO term "anterior/posterior specification". Taken together, we find that scPCA effectively handles scRNA-seq datasets with differences in cell type distribution across conditions.

## Discussion

We introduced scPCA, a factorization model designed to analyze multi-condition single-cell data. By informing our factor model about the conditioning variable, we define components for each condition relative to a chosen reference. This approach enables the model to establish distinct but related coordinate systems, effectively addressing shifts in gene expression patterns caused by experimental interventions. As a result, scPCA generates integrated cell embeddings and provides interpretable gene loading weights. Moreover, by calculating the difference in gene loading weights across conditions, the model highlights factor-specific patterns of upregulated or downregulated gene expression.

We applied our method across a range of single-cell datasets, showcasing the effectiveness of scPCA in handling a variety of scenarios. This included modeling both stark and modest perturbation effects, multiple time points, datasets characterized by differences in cell type distribution across conditions, and integrating batch effects. We anticipate that the linear model of scPCA will be instrumental in integrating and interpreting multi-condition single-cell datasets. Looking ahead, we see strong potential for extending this framework to multi-omic settings, where integrating transcriptomic, epigenomic, or proteomic measurements from the same cells could provide a more comprehensive view of how biological systems respond to perturbations.

Despite its strengths, scPCA has some limitations. A key constraint is its assumption that the principal axes of variation are largely conserved across conditions. Because scPCA expresses the bases of each condition as a linear function of the reference basis/condition, the model primarily captures linear expression changes. This means it may struggle in scenarios where gene expression changes are non-linear. Additionally, scPCA is best positioned as a tool for exploratory analysis. While loading weight differences often highlight up- or down-regulated genes, these findings should be further validated through complementary methods. Lastly, our approach uses approximate Bayesian inference to estimate the posterior distribution of parameter weights, which can lead to an underestimation of their variance, potentially affecting the precision of these estimates.

## Methods
### The scPCA model

For a complete derivation of the model, see the Supplementary Methods. The scPCA factorization requires a multi-condition single-cell expression count matrix $X \in \mathbb{N}_0^{C \times G}$ for $C$ cells and $G$ genes, a design matrix $D \in \{0,1\}^{C \times Q}$ that encodes a conditioning variable with $Q$ levels, and an optional indicator matrix $B \in \{0,1\}^{C \times P}$ that may adjust for condition-specific mean gene expression for $P$ groups of cells ($P = 1$ or $P = Q$, i.e., scPCA fits a global or condition-specific mean offset(s), respectively) as inputs. We model the elements of the expression count matrix $X = \{x_{cg}\}$ for cell $c = 1, \ldots, C$ and gene $g = 1, \ldots, G$ as negative binomial distributed[27], given an unobserved gene expression level $\mu_{cg}$ and a gene-specific over-dispersion $\alpha_g$

$$x_{cg} \sim \mathrm{NB}\left(\mu_{cg}, \alpha_g\right) \tag{1}$$

In scPCA, we decompose the unobserved gene expression as a linear function of the conditioning variable encoded in $D$. Denote the entries of the design and indicator matrix $D = d_{cq}$ and $B = \{b_{cp}\}$ for level $q = 1, \ldots, Q$

and group $p = 1, \ldots, P$, respectively, and let $l_c = \log \sum_{g=1}^{G} x_{cg}$ denote a cell-specific fixed size factor (total number of counts). Then,

$$\log \mu_{cg} = \underbrace{l_c}_{\text{Size factor}} + \underbrace{\left( \sum_{p=1}^{P} b_{cp} v_{pg} \right)_{cg}}_{\text{Intercept term}} + \underbrace{\left( \sum_{q=1}^{Q} d_{cq} \cdot \sum_{f=1}^{F} z_{cf} w_{fgq} \right)_{cg}}_{\text{Factor model}} \quad (2)$$

where $w_{fgq}$ represents the entries of a loading weight tensor $W \in \mathbb{R}^{F \times G \times Q}$ consisting of $f = 1, \ldots F$ components, $z_{cf}$ the elements of a factor weight matrix $Z \in \mathbb{R}^{C \times F}$ containing the coordinates of each cell with respect to $W$, and $v_{pg}$ the entries of a mean offset $V \in \mathbb{R}^{P \times G}$ matrix.

## Interpretation of the factors *Z*
Similar to interpreting factors in PCA, scPCA factors denote lower-dimensional representations of global sources of variability in the data. Each factor arranges cells along a principal axis of variation centered at zero. Cells with positive or negative factor weights manifest opposite phenotypes, with the magnitude of the factor weight indicating the strength of the effect.

## Interpretation of the loading weights *W*
The loading weights provide a score for the importance of each gene to its corresponding factor. Loading weights close to zero indicate genes with little or no importance to the factor, whereas genes with large absolute values exhibit a strong factor association. The direction of the effect is indicated by the sign of the weight: a positive weight suggests higher gene expression in cells with positive factor values, and conversely for negative weights.

Central to scPCA's modeling approach is its ability to infer loading weight differences (LWDs) across the levels of the conditioning variables. This enables an assessment of gene expression shifts along each component of the decomposition. To accomplish this, scPCA relies on the user to establish a reference level for comparison, and encodes the categorical conditioning variables in terms of a design matrix. For example, in the common two-class comparison setting (e.g., control versus treated), $D$ is a $C \times 2$ design matrix with a column of ones corresponding to an intercept and a column of indicator variables for the class of each cell (e.g., 0 for control and 1 for treated). Here, the intercept column defines the reference level, and instructs the model to fit the basis of the treatment condition as an offset of the basis of the control condition (for detailed derivation, please refer to the Supplementary Methods).

## Note on the mean offset parameter *V*
In scPCA, we directly model single-cell expression data in count space, thereby evading the need to transform the data prior to the analysis. However, this prohibits to mean-center the data, which is an obligatory preprocessing step for PCA. For this reason, we introduce an explicit parameter to capture either the global or condition-specific mean expression of each gene. Empirically, we observed that for most single-cell datasets, a global mean offset is sufficient (in this case, $B$ reduces to a vector of ones). However, additional adjustments may be necessary in cases when the conditioning variable affects expression globally.

## Model inference
We employ stochastic variational inference to estimate the model parameters and implement scPCA in the probabilistic programming language Pyro[23], which supports GPU acceleration. As with many probabilistic latent variable models, the underlying optimization problem is non-convex, meaning the algorithm may converge to different local optima depending on initialization. Consequently, scPCA produces non-deterministic outputs, and parameter estimates may vary slightly across runs. To address this, we recommend users perform multiple initializations and select the model yielding the strongest evidence lower bound (ELBO)—a standard criterion for selecting the best variational approximation. Full details of the inference

procedure and model implementation are provided in the Supplementary Methods.

## Model selection
Determining the appropriate number of factors is challenging: too few factors might miss important biological insights, while too many factors can cause an "oversegmentation" of the data, thus complicating interpretation. To find the appropriate decomposition rank for a given dataset, we propose to employ the root mean squared error (RMSE) between observed and reconstructed expression values. In our simulation experiments, we found a sharp flattening of the RMSE curve at the true rank of the data (Supplementary Fig. 1a), indicating that adding more components does not substantially improve model fit.

However, in our experiments, we found that for most mid-sized single-cell datasets, 20 factors provided a good balance between complexity and interpretability. Moderate changes in the number of factors did not change results substantially, most likely due to the fact that the first components usually explain most of the variance in the data. We note that scPCA returns its factors by importance, i.e., ranked by the variance explained of each individual factor.

## Simulation experiments
For the simulation experiments, we simulated data according to the generative model (Supplementary Methods) using $F = 10$ factors, $G = 2000$ genes, and $Q = 2$ conditions. To assess the scPCA inference under various settings, we varied the number of cells per condition $C_q \in \{10, 100, 1000, 10000\}$ and used different levels of negative binomial and Poisson noise to simulate the data. To evaluate the fit of the model, we calculated the root mean square error (RMSE) between the predicted counts of the fitted model and the simulated ground truth.

## Subspace reconstruction
Recall that different sets of basis vectors $\{w_1, \ldots, w_F\}$ and $\{v_1, \ldots, v_F\}$ may span the same vector space. More formally, if $W$ and $V$ consist of the column vectors $\{\boldsymbol{w}_1, \ldots, \boldsymbol{w}_F\}$ and $\{\boldsymbol{v}_1, \ldots, \boldsymbol{v}_F\}$, respectively, and $x = Wy$, then there exists a vector $z$ such that $\boldsymbol{z} = \boldsymbol{wz}$. To see this, consider first the case when $W$ and $V$ are of full rank, then $W$ and $V$ are both invertible. Then

$$x = Wy = VV^{-1}Wy = Vz$$

where $z = V^{-1}Wy$.

However, since scPCA decompositions are generally not full rank (i.e., $F \ll G$) we instead consider the projection matrix of the estimated loading tensor $\hat{W}$,

$$P = \hat{W} (\hat{W}^T \hat{W})^{-1} \hat{W}$$

If $\hat{W}$ and the simulated ground truth $W$ span the same subspace, then $PW = W$. Thus, to evaluate how accurately scPCA reconstructs the data-generating subspaces, we compute the RMSE between $PW$ and $W$.

## Single-cell data preprocessing and analysis
We downloaded the filtered raw-count single-cell data from Zenodo [17], GSE124872 [33], GSE102827 [41], 10xGenomics [32], Figshare [18], and GSE112294 [44]. We filtered genes using highly variable gene selection (HVG) using scanpy's "seurat_v3" setting to reduce the number of total genes to 4000[46]. We then applied the logCPM10k transformation to the count data before subjecting them to a rank 20 PCA decomposition. For the scPCA analysis, we performed an equivalent rank 20 decomposition on the raw count data.

## Gene set enrichment analysis
To associate scPCA factors with biological processes, we used the gseapy[47] Python package (version 1.0.4) using either Reactome or Gene Ontology gene sets. We used the top 200–300 genes with the highest or lowest loadings

and performed gene set overrepresentation analysis using all genes in the data set as background.

## Reporting summary

Further information on research design is available in the Nature Portfolio Reporting Summary linked to this article.

## Data availability

The datasets used in this work may be downloaded from Zenodo[17,48], GSE124872[33], GSE102827[41], 10xGenomics[32], Figshare[18,49], and GSE112294[44].

## Code availability

The scPCA software package is available on PyPI and Zenodo[50]. The source code and documentation are available on GitHub. The source data for the figures of this paper are provided in Supplementary Data files 1 and 2.

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

## Acknowledgements

We would like to thank Wolfgang Huber, Tumay Capraz, Stefan Pleidli, Constantin Ahlman-Eltze, and Petr Smirnov for useful discussions and comments on the paper. We also thank Britta Velten, Martin Rohbeck, and Kai Ueltzhöffer for insightful comments on the manuscript.

## Author contributions

H.V. conceived and implemented the scPCA software. H.V wrote the paper and designed the figures.

## Funding

## Competing interests

The authors declare no competing interests.
