## [Transparent Peer Review file · Communications Biology]

Joint Modeling of Cellular Heterogeneity and Condition Effects with scPCA in Single-Cell RNA-Seq

Corresponding Author: Dr Harald Vöhringer

Version 0:

Reviewer comments:

Reviewer #1

(Remarks to the Author)

The authors propose a method known as sc (single-cell)PCA, which can jointly model cellular heterogeneity and condition effects (e.g., batch effect, perturbations) for single-cell transcriptomics data under different conditions. The authors consider various different cases and demonstrate the applicability of scPCA as a general tool. However, I have some questions and concerns that might preclude the publication of this paper in its current version.

1. Motivation: Handling batch effect and perturbation effect are separate problems. That is because we expect to integrate data with different batches by mixing the cells with the same cell types across batches. However, for perturbation problems, cells from the same cell type may still contain different biological variations. Therefore, how do the authors consider these two different settings based on the same pipeline? It is hard to convince people that scPCA can handle both perturbation modeling and batch effect reduction.
2. This paper lacks important benchmarking analysis with different methods for both batch effect correction and perturbation analysis. For example, scVI (<https://www.nature.com/articles/s41592-018-0229-2>), ResPAN (<https://pubmed.ncbi.nlm.nih.gov/35771600/>), and Harmony should be included in the benchmarking analysis, as scPCA is also based on deep-learning design, and thus it is essential to compare models with similar settings or well-known good performances. For perturbation analysis, CINEMA-OT (<https://www.nature.com/articles/s41592-023-02040-5>) and CellANOVA (<https://www.nature.com/articles/s41587-024-02463-1>) should also be taken into consideration for benchmarking analysis.
3. It is also interesting to compare this paper's proposed method with the other scPCA (sparse and contrastive PCA, <https://academic.oup.com/bioinformatics/article/36/11/3422/5807607>), which tries to address a similar problem.
4. Analyzing single-cell multi-omic data is an interesting topic, as there exist a couple of methods for modeling transcriptomics only, so the novelty is not impressive. Is it possible to extend scPCA for multi-omic data analysis (e.g., 10X Multiome PBMC data)? If not, could the authors offer any potential solutions for future extensions?
5. In the paragraph of line 359, the authors mention that the model varies slightly with different initializations. How different are they? What is the meaning of "try different initialization and select the model with best ELBO score"? Is it a fair design? Also, did the authors check if their model performance will not be affected by random seeds and thus the results can be reproduced and not biased by random seeds? The others might also consider correct some typos, for example, it should be "the best ELBO score" in the same sentence.
6. The method lacks comparison with PCA or other baselines based on efficiency, e.g., memory usage and training time are important factors to present.
7. The authors select multiple datasets for analysis, are they just "cherry-pick" examples? Could the authors provide proofs to demonstrate that these datasets are representative of certain problems? For example, how to trust the results after batch effect correction based on a pure-biological-variation-affected dataset? The authors might consider some metrics to quantify the proportion of batch variation and biological variation for a given dataset.
8. The codes of this paper are well-organized and scPCA can be installed smoothly. I encourage the authors to include more examples in the tutorial part and let users know the contributions of scPCA at a better level.
9. On minor comment: The figure organization and panel organization are hard to read and need to be improved. For example, in Figure 2-4 (a), the legends overlap with panels, which makes me feel hard to recognize cells. Also, each figure should obtain one page for presentation, the authors might consider improving it.

Reviewer #2

(Remarks to the Author)

The paper presents scPCA, a factorization model for analyzing multi-condition scRNA-seq data by incorporating conditioning variables into a linear dimensionality reduction framework. I have several concerns:

1. How does scPCA compare with leading integration methods like Harmony, Seurat? Would a benchmarking experiment including runtime, scalability, and clustering accuracy enhance the evaluation?
2. The manuscript assumes that principal axes of variation are largely conserved across conditions. How does this affect datasets with highly non-linear relationships? A comparison with non-linear methods like a vanilla autoencoders could help assess potential biases.
3. Why was an overdispersed Negative Binomial distribution selected for modeling count data instead of more commonly used ZINB model for scRNA-seq data?
4. Were different decomposition ranks systematically tested to determine the optimal number of factors? Is there guidance for users on selecting parameters for different datasets? Were the same hyperparameters applied across all datasets? I could not find them in the manuscript.

Reviewer #3

(Remarks to the Author)

The authors present a flexible factorization model, scPCA, for analyzing multi-condition single-cell RNA-seq data. This model allows for the assessment of expression shifts and the extraction of biological signals across conditions. The performance of scPCA is evaluated using both simulated and real scRNA-seq datasets, specifically in terms of its ability to model batch effects, extract cell-type-specific factors, and handle multiple time-point data, among other aspects. Below are my comments on the current manuscript version.

Major Comments:

1. Although the model incorporates categorical covariates representing conditions (e.g., drug treatment or genetic perturbation), it currently only considers the condition mean. The distinction between batch effects and true biological signals remains unclear. My concern is how the model differentiates these two factors. Section 2 of the Results demonstrates that scPCA can model batch effects and shows good performance, but the interpretation of these results is not fully explained. How can we confidently distinguish batch effects from biological signals in the model?
2. For the interferon-beta stimulation dataset in PBMCs, we would expect to see results where one or two cell types, such as CD14-positive monocytes, are clearly associated with IFN-beta stimulation. These results should ideally be visualized in a UMAP plot (e.g., Figures 2b or 2e), where perturbation-associated cell types should show distinct separation. However, some of the perturbation-associated cell types do not appear well-aligned in the UMAP, which needs further clarification.
3. How are the LWDs (Loading Weight Differences) computed in the model? Please provide more detail on this process.
4. In the simulations, the authors should demonstrate scPCA's ability to model batch effects, extract cell-type-specific factors, etc. Additionally, I am concerned about the use of root mean squared error (RMSE) between PW and W in the loss function. Since PW and W come from two different spaces, they may represent different variables, which raises the question of how appropriate it is to compute RMSE between them. These variables may differ substantially but still be reasonable in the context of their respective spaces.
5. The authors should include a comparison with CellANOVA (Zhaojun Zhang et al., 2024, Nature Biotechnology) to provide a broader context for the performance of scPCA.
6. How should one determine the optimal number of scPCs for analysis in real datasets? The methodology for selecting the number of principal components is not sufficiently addressed.

Version 1:

Reviewer comments:

Reviewer #1

(Remarks to the Author)

I sincerely thank the authors for making this comprehensive revision. However, I still have some concerns about result justification and necessary benchmarking analysis.

1. The authors did not compare scPCA with other baselines for batch effect correction in the quantitative approach. Only one score like iLISI is not a strong proof and it seems that Seurat also did well in all of the four datasets. I would encourage the

authors take a look at scIB and include other metrics to reflect the evaluation from both batch and bio perspective.

Also, could the authors explain why methods such as Harmony show imbalanced performances across other cases, since Harmony is a top-performer in various benchmarking analysis, does this mean Harmony works for real data? I think it is necessary to include one or two real data to justify their contributions.

2. I understand CINEMA-OT cannot be included in this research, but I did not find reasons for not including CellANOVA here. I do not believe that CellANOVA will be much affected by the results from Harmony as it is designed to tackle the missing covariate problems caused by Harmony. I think it is necessary to include CellANOVA in the benchmarking analysis.

Reviewer #2

(Remarks to the Author)

The authors have addressed all of my major concerns. Before publication, I have one minor suggestion: It is commendable that the authors included a fair discussion on the limitations of scPCA in capturing non-linear changes. Supplementary Figure 7 is particularly important in demonstrating the performance of the proposed method, and I recommend that it be included in the main manuscript.

Reviewer #3

(Remarks to the Author)

No further comments on it.

Version 2:

Reviewer comments:

Reviewer #1

(Remarks to the Author)

Thank you for your great work. I will support the publication of this paper.

Reviewer 1

The authors propose a method known as *sc* (single-cell)PCA, which can jointly model cellular heterogeneity and condition effects (e.g., batch effect, perturbations) for single-cell transcriptomics data under different conditions. The authors consider various different cases and demonstrate the applicability of *sc*PCA as a general tool. However, I have some questions and concerns that might preclude the publication of this paper in its current version.

1. Motivation: Handling batch effect and perturbation effect are separate problems. That is because we expect to integrate data with different batches by mixing the cells with the same cell types across batches. However, for perturbation problems, cells from the same cell type may still contain different biological variations. Therefore, how do the authors consider these two different settings based on the same pipeline? It is hard to convince people that *sc*PCA can handle both perturbation modeling and batch effect reduction.

We thank the reviewer for this thoughtful and important comment. We fully agree that batch effect correction and perturbation modeling are conceptually distinct challenges: batch effects arise from technical variability and should be removed to facilitate data integration, while perturbations reflect meaningful biological variation that should be preserved and interpreted.

To successfully disentangle these two sources of variation, **scPCA relies on a balanced experimental design** — that is, the dataset includes all combinations of batch and perturbation conditions. This structure is essential to ensure that the model can distinguish between technical and biological effects in a statistically identifiable way.

To clarify how *sc*PCA addresses this challenge, we have added two illustrative examples in the **Supplementary Methods** under the section “*Disentangling Batch and Condition Effects*“. These simulations on 2-dimensional synthetic datasets demonstrate the mathematical conditions under which *sc*PCA successfully separates batch effects from biological perturbations.

“The flexible factorization framework of *sc*PCA enables it to disentangle batch effects from biological conditions in *sc*RNA-seq data. However, to successfully separate batch and perturbation effects, *sc*PCA requires a balanced experimental design—that is, all combinations of batch and perturbation must be represented. This design is crucial to ensure the model can statistically identify and distinguish technical variation from true biological signals.

To exemplify how *sc*PCA addresses this challenge, we discuss two illustrative examples using a simulated 2-dimensional toy dataset. These examples demonstrate intuitively how *sc*PCA disentangles batch and perturbation effects under two different assumptions:

1. In the first scenario, **batch effects manifest as global shifts in mean expression**, while **perturbations correspond to rotations in latent space**.
2. In the second scenario, both **batch and perturbation effects arise from rotations**.

While these examples focus on shifts in location and rotations, we note that the *sc*PCA framework is, in principle, capable of capturing a broad range of affine

transformations—including shearing and scaling—through its condition- and batch-specific parameterization of the loading tensor \mathbf{W} and the mean offset matrix \mathbf{V} .

In the first example, perturbations are represented as rotations (P) in data space, while batch effects result from shifts in location. In real single-cell datasets, such batch effects often present as global shifts in average gene expression, for example due to differences in sequencing protocols or platforms. scPCA addresses this by allowing the user to specify a mean offset matrix \mathbf{V} , which accounts for batch-specific shifts. This enables the model to recover a shared low-dimensional structure (right panel) while simultaneously learning appropriate transformations to align perturbed and reference data (left panel). As a result, scPCA captures consistent biological variation while preserving underlying cell-type structure across batches.

The second example represents a more complex scenario in which both batch and perturbation effects arise from rotations. To disentangle these sources of variation, scPCA relies on an appropriate loading design formula, such as $\text{batch} + \text{design}$, which instructs the model to represent each batch–condition combination with a distinct linear combination of the loading tensor. This formulation allows scPCA to accurately capture the shared perturbation direction (encoded in the loading tensor slice \mathbf{W}^3) while learning batch-specific linear transformations (loading tensor slice \mathbf{W}^2). As a result, the model successfully separates biological perturbation effects from batch-induced distortions in the data space.

We thank the reviewer for the critical reading of our manuscript and providing insightful comments. We believe that their comments and suggestions have greatly strengthened our revised manuscript.

2. *This paper lacks important benchmarking analysis with different methods for both batch effect correction and perturbation analysis. For example, scVI (<https://www.nature.com/articles/s41592-018-0229-2>), ResPAN (<https://pubmed.ncbi.nlm.nih.gov/35771600/>), and Harmony should be included in the benchmarking analysis, as scPCA is also based on deep-learning design, and thus it is essential to compare models with similar settings or well-known good performances. For perturbation analysis, CINEMA-OT (<https://www.nature.com/articles/s41592-023-02040-5>) and CellANOVA (<https://www.nature.com/articles/s41587-024-02463-1>) should also be taken into consideration for benchmarking analysis.*

We thank the reviewer for their thoughtful and constructive comments on the benchmarking analyses. We fully agree that it is essential to compare scPCA to established methods for batch correction and perturbation modeling, particularly those that rely on latent variable models.

To address this, we conducted a comprehensive benchmarking experiment comparing scPCA with PCA, Harmony, Seurat (CCA), and scVI, as summarized in **Supplementary Figure 7** of the revised manuscript. We also updated the **Results** section accordingly to reflect these additions:

*“... We further benchmarked scPCA against other single-cell latent variable models (**Supplementary Fig. 7a-c**) and found that it maintains competitive integration performance across a range of scenarios. Notably, scPCA scales efficiently with increasing dataset size, exhibiting competitive runtime and memory usage compared to other methods^{5,6,15} (**Supplementary Fig. 7d**).”*

We simulated data according to the generative process underlying scPCA (**see Supplementary Methods Eq. 4**), evaluating performance across both a **“simple”** and a **“complex”** integration task involving one or two cell types across two batches.

In these benchmarking experiments, scPCA demonstrates consistently strong performance across all integration tasks. In the “simple” integration task, scPCA successfully removes batch effects across all dataset sizes (100 to 100,000 cells), and preserve the homogeneous cell population, yielding well-integrated representations comparable to those from Harmony, scVI and Seurat. This confirms that scPCA does not introduce spurious structure when variation is limited (**Supplementary Fig. 7a**). For the more challenging “complex” simulation, scPCA effectively corrects for batch while maintaining cell type separation. In contrast, PCA, Harmony and scVI exhibit batch effects, particularly at smaller cell counts. Seurat performs well across the assessed range of total cell counts but shows minor distortions at small dataset sizes (**Supplementary Fig. 7b**).

In addition to the qualitative assessment using UMAP embeddings, we quantitatively evaluated integration performance using the iLISI metric for the complex simulation scenario. The results are consistent with the visual assessment: scPCA corrects for batch effects across all dataset sizes. In contrast, PCA, Harmony, and scVI show diminished integration quality, particularly at smaller sample sizes. Seurat performs well overall but exhibits slight distortions in low-cell-count settings (Supplementary Fig. 7c).

We also evaluated the **computational efficiency of scPCA** in comparison to the benchmarked methods above by measuring CPU time and memory usage across increasing dataset sizes under both the simple and complex simulation scenarios. As shown in Supplementary Figure 7d, scPCA demonstrates favorable scaling behavior, maintaining low CPU time and memory consumption relative to more computationally intensive methods such as Seurat and scVI. Notably, scPCA consistently requires substantially less memory than Seurat, which exceeds 40 GB on the largest datasets. Additionally, scPCA completes in under 500 seconds even for 100,000 cells, whereas Seurat and scVI require significantly more time, with Seurat exceeding 1,500 seconds in the complex task.

Regarding the specific suggestions:

- ResPAN was not included, as its architecture and goals are closely aligned with those of scVI, which we did include. Since both are deep generative models for batch correction, we consider scVI a sufficient representative of this class of models.
- CINEMA-OT, while an excellent tool for perturbation analysis, does not learn a shared latent representation across samples. As scPCA is fundamentally a latent variable model, CINEMA-OT falls outside the scope of our latent-factor benchmarking.
- CellANOVA relies on a latent representation derived via Harmony, which is included in our benchmark. Since CellANOVA's performance critically depends on Harmony's integration step, we believe that including Harmony already provides a relevant reference.

3. It is also interesting to compare this paper's proposed method with the other scPCA (sparse and contrastive PCA, <https://academic.oup.com/bioinformatics/article/36/11/3422/5807607>), which tries to address a similar problem.

We thank the reviewer for this helpful suggestion. While both sparse contrastive PCA and scPCA aim to distinguish biological signal from background variation, they differ in formulation, scope, and output. Technically, sparse contrastive PCA finds a projection that emphasizes differences between datasets, whereas our method returns a reference basis along with condition-specific perturbation vectors that describe how each basis component is altered by a given condition.

We have now clarified this distinction in the Supplementary Methods section and now cite Boileau et al. to acknowledge their contribution and avoid confusion between the two frameworks.

“Sparse contrastive PCA is a deterministic, linear dimensionality reduction method designed to identify directions of variation that are enriched in a target dataset relative to a background dataset. It does so by optimizing a contrastive objective while enforcing sparsity in the loading vectors, thereby emphasizing features that distinguish the target from the background.

In contrast, scPCA takes a probabilistic, generative modeling approach. It learns a shared low-dimensional representation from reference (unperturbed) data and models how this basis is altered across experimental conditions through condition-specific linear transformations. Rather than directly contrasting target and background datasets, our scPCA explicitly captures shared variation and characterizes how each latent component shifts under different perturbations, providing a factor-wise view of biological variation across conditions.”

We appreciate the reviewer bringing this to our attention.

4. Analyzing single-cell multi-omic data is an interesting topic, as there exist a couple of methods for modeling transcriptomics only, so the novelty is not impressive. Is it possible to extend scPCA for multi-omic data analysis (e.g., 10X Multiome PBMC data)? If not, could the authors offer any potential solutions for future extensions?

We thank the reviewer for raising this important point. While scPCA was originally developed for transcriptomic data, its underlying generative modeling framework is flexible and, in principle, could be extended to multi-omic settings.

A potential direction would be to jointly model multiple modalities—such as gene expression and chromatin accessibility—by incorporating modality-specific observation models while sharing a common latent structure. This approach could build on concepts from multi-view latent variable models (e.g. MOFA) to disentangle shared and modality-specific variation.

However, we note that such an extension would require substantial methodological development, including changes to the model architecture and data preprocessing pipelines. As such, this would constitute a new project beyond the scope of the current manuscript. We have now clarified this point in the **Discussion** and highlighted the integration of multi-omic data as a promising direction for future work.

“...We anticipate that the linear model of scPCA will be instrumental in integrating and interpreting multi-condition single-cell datasets. Looking ahead, we see strong potential for extending this framework to multi-omic settings, where integrating transcriptomic, epigenomic, or proteomic measurements from the same cells could provide a more comprehensive view of how biological systems respond to perturbations.”

We appreciate the reviewer’s insightful suggestion, which aligns well with our vision for future developments of scPCA.

5. In the paragraph of line 359, the authors mention that the model varies slightly with different initializations. How different are they? What is the meaning of “try different

initialization and select the model with best ELBO score”? Is it a fair design? Also, did the authors check if their model performance will not be affected by random seeds and thus the results can be reproduced and not biased by random seeds? The others might also consider correct some typos, for example, it should be “the best ELBO score” in the same sentence.

We thank the reviewer for these valuable observations. Regarding model initialization: as with many variational inference frameworks, **scPCA's optimization landscape is non-convex**, and therefore **different initializations can lead to slightly different local optima**. In our experiments, the variations between runs are typically minor and do not qualitatively affect the inferred latent space or the conclusions drawn from the data. Quantitatively, the variation in ELBO scores across different runs is small, and we observe consistent biological signal recovery (see attached Figure in which we show 10 scPCA runs).

When we state that we “try different initializations and select the model with the best ELBO score,” we mean that we train the model multiple times with different random seeds and select the run that achieves the highest evidence lower bound (ELBO) on the training data. This practice is standard and principled in variational inference, as the ELBO provides a lower bound on model likelihood and reflects the fit of the model to the data.

We have now clarified this point in the revised manuscript:

“... acceleration. As with many probabilistic latent variable models, the underlying optimization problem is non-convex, meaning the algorithm may converge to different local optima depending on initialization. Consequently, scPCA produces non-deterministic outputs, and parameter estimates may vary slightly across runs. To address this, we recommend users perform multiple initializations and select the model yielding the highest evidence lower bound (ELBO)—a standard criterion for selecting the best variational approximation. Full details of the inference procedure and model implementation are provided in the Supplementary Methods.”

We appreciate the reviewer’s attention to detail and helpful suggestions.

6. The method lacks comparison with PCA or other baselines based on efficiency, e.g., memory usage and training time are important factors to present.

We thank the reviewer for this valuable suggestion. We have addressed this point in our response to **Point 1**, where we now include a comprehensive benchmarking analysis comparing scPCA to PCA and other baseline methods (Harmony, Seurat, scVI) in terms of efficiency metrics such as memory usage and training time. As shown in **Supplementary Figure 7d**, scPCA demonstrates favorable scalability and competitive runtime performance across increasing dataset sizes.

7. The authors select multiple datasets for analysis, are they just “cherry-pick” examples? Could the authors provide proofs to demonstrate that these datasets are representative of certain problems? For example, how to trust the results after batch effect correction based on a pure-biological-variation-affected dataset? The authors might consider some metrics to quantify the proportion of batch variation and biological variation for a given dataset.

We thank the reviewer for raising this important point. We agree that careful dataset selection is critical to avoid biased evaluations and to ensure that methods are assessed under representative and challenging conditions.

To this end, we have evaluated scPCA across a diverse set of datasets and scenarios, which are systematically summarized in Extended Data Figures 1–6. These datasets were not cherry-picked, but rather chosen to represent a range of common challenges in single-cell data analysis, including platform heterogeneity, donor variability, and nested batch effects. This broad evaluation allows us to assess the strengths and limitations of scPCA under varying degrees of technical and biological confounding.

For instance:

- Extended Data Fig. 1 shows that scPCA successfully integrates data across different technologies (1b, d).
- Extended Data Fig. 2 explores integration across multiple donors for a homogeneous cell population. While scPCA produces a visually coherent UMAP embedding (2a), it yields lower batch scores compared to PCA (2d), highlighting trade-offs between batch removal and biological fidelity.
- In Extended Data Fig. 3 and 4, we analyze complex datasets from Luecken et al., where samples vary by spatial location, lab, protocol, and species. These scenarios expose the limitations of scPCA's linear framework, as full batch correction is not achieved.
- Extended Data Fig. 5 and 6 involve synthetic datasets with either variable cellular composition or nested batch effects. scPCA performs well when cellular composition varies, but struggles when batch effects are hierarchically structured.

Taken together, these analyses provide a systematic and balanced assessment of scPCA's capabilities. We acknowledge the value of quantifying batch vs. biological variation and agree this remains an active area of research. Our findings suggest that scPCA is well-suited for many realistic integration tasks, especially where linear batch effects dominate, but may be limited in non-linear scenarios.

8. The codes of this paper are well-organized and scPCA can be installed smoothly. I encourage the authors to include more examples in the tutorial part and let users know the contributions of scPCA at a better level.

We thank the reviewer for their positive feedback and helpful suggestion. In response, we have added two comprehensive tutorials that guide users through key applications of scPCA. These tutorials include examples directly from the manuscript, such as the analysis of a complex single-cell dataset of lung cells from aging mice comprising more than 20 distinct cell types (<https://sagar87.github.io/scPCA/notebooks/angelidis.html>), and a dataset of brain cells from light-stimulated mice encompassing multiple experimental conditions (<https://sagar87.github.io/scPCA/notebooks/hrvatin.html>). We believe these additions will help users better understand the functionality and contributions of scPCA, and facilitate its application to a broad range of single-cell studies.

9. On minor comment: The figure organization and panel organization are hard to read and need to be improved. For example, in Figure 2-4 (a), the legends overlap with panels, which makes me feel hard to recognize cells. Also, each figure should obtain one page for presentation, the authors might consider improving it.

We thank the reviewer for this helpful comment. We have revised the figure layout and improved panel organization to enhance clarity and readability. We hope these changes improve the overall presentation and ease of interpretation.

Reviewer #2 (Remarks to the Author):

The paper presents scPCA, a factorization model for analyzing multi-condition scRNA-seq data by incorporating conditioning variables into a linear dimensionality reduction framework. I have several concerns:

1. How does scPCA compare with leading integration methods like Harmony, Seurat? Would a benchmarking experiment including runtime, scalability, and clustering accuracy enhance the evaluation?

We thank the reviewer for raising this important point. As noted in our response to **Reviewer 1** above, we have conducted a comprehensive benchmarking analysis comparing scPCA with leading integration methods, including PCA, Harmony, Seurat, and scVI. This evaluation includes clustering accuracy, batch correction performance, runtime, and memory usage across a range of dataset sizes (from 100 to 100,000 cells).

These results—summarized in **Supplementary Figures 7**—demonstrate that scPCA performs competitively in terms of integration quality while offering strong scalability and efficient resource usage. We have updated the **Results** and **Supplementary Materials** accordingly, and thank the reviewer for this helpful suggestion.

2. The manuscript assumes that principal axes of variation are largely conserved across conditions. How does this affect datasets with highly non-linear relationships? A comparison with non-linear methods like a vanilla autoencoders could help assess potential biases.

We thank the reviewer for this insightful comment. scPCA indeed assumes that **principal axes of variation are approximately linear and conserved across conditions**, which suits many multi-condition single-cell datasets.

However, in more complex datasets with confounding factors like spatial sampling, protocol differences, or cross-species comparisons, variation may not be linear. In such cases, scPCA's linear framework may struggle to separate technical from biological variation. This limitation is demonstrated in **Extended Data Figures 3 and 4**, where scPCA has difficulty integrating data with layered or heterogeneous batch effects despite good performance in simpler settings. Additionally, our benchmarks now include scVI, demonstrating how an autoencoder performs on data generated under the (linear) scPCA model (**Supplementary Fig. 7**). As noted in the **Discussion**, scPCA may have troubles capturing non-linear changes.

To systematically assess how non-linearities affect results in (linear) factor models compared to vanilla autoencoders would require substantial additional work and is beyond the scope of this study, likely warranting a separate publication. Future extensions incorporating non-linear approaches could help overcome these challenges.

3. Why was an overdispersed Negative Binomial distribution selected for modeling count data instead of more commonly used ZINB model for scRNA-seq data?

We thank the reviewer for this insightful question. We chose to model scRNA-seq count data using an overdispersed Negative Binomial (NB) distribution rather than a Zero-Inflated Negative Binomial (ZINB) for several reasons:

First, recent studies—including Svensson (Svensson 2020)—have shown that zero-inflation is not necessary for droplet-based scRNA-seq data. The apparent excess of zeros can be sufficiently explained by the overdispersion of the NB distribution, particularly when biological variability is appropriately modeled.

Second, the Negative Binomial (NB) distribution provides a simpler and more tractable generative model, while still capturing key properties of count data—such as the mean–variance relationship. In contrast, Zero-Inflated Negative Binomial (ZINB) models introduce an additional Bernoulli component to account for excess zeros, which adds discrete random variables that complicate gradient-based optimization. Given that scalability is a critical requirement for single-cell data analysis, we chose the NB distribution to enable more efficient and stable training.

We have clarified this modeling choice in the revised **Supplementary Methods** section and now explicitly cite relevant literature supporting the use of the NB distribution for modeling single-cell RNA-seq count data.

“Importantly, scPCA models the observed count data using a Negative Binomial (NB) likelihood. We opted for the NB distribution over a zero-inflated NB model because recent studies indicate that zero-inflation is unnecessary for droplet-based scRNA-seq data². The NB distribution effectively captures essential count properties such as overdispersion and the mean–variance relationship, while avoiding the added complexity of zero-inflated

models. This choice enables more efficient and stable gradient-based optimization, which is crucial for scalable single-cell data analysis.”

4. Were different decomposition ranks systematically tested to determine the optimal number of factors? Is there guidance for users on selecting parameters for different datasets? Were the same hyperparameters applied across all datasets? I could not find them in the manuscript.

We thank the reviewer for this helpful question. Yes, we systematically evaluated different decomposition ranks to determine the optimal number of latent factors. As shown in **Supplementary Figure 1a**, we used the root mean squared error (RMSE) between the observed and reconstructed expression values as a quantitative model selection criterion. In our experiments, we observed that the RMSE curve saturates at the true decomposition rank, indicating that beyond a certain number of factors, additional components do not improve model fit. Importantly, we found that the results are fairly robust to moderate changes in the number of factors. The learned factorisation and biological interpretation remain stable within a reasonable range of ranks, giving users flexibility without requiring precise tuning.

We also explored the use of information criteria such as AIC and BIC, but found these to be less reliable in the context of variational inference and high-dimensional single-cell data. These criteria tended to either over- or under-estimate the appropriate rank in simulation experiments, likely due to their reliance on asymptotic assumptions that are difficult to satisfy in this setting.

To support practical use, we have updated the Methods section to clarify this procedure and now include guidance in the software documentation for selecting decomposition ranks using RMSE-based validation.

“Determining the appropriate number of factors is challenging: too few factors might miss important biological insights, while too many factors can cause an “oversegmentation” of the data, thus complicating interpretation. To find the appropriate decomposition rank for a given dataset, we propose to employ the root mean squared error (RMSE) between observed and reconstructed expression values. In our simulation experiments, we found a sharp flattening of the RMSE curve at the true rank of the data (Supplementary Fig. 1a), indicating that adding more components does not substantially improve model fit.”

Reviewer #3 (Remarks to the Author):

The authors present a flexible factorization model, scPCA, for analyzing multi-condition single-cell RNA-seq data. This model allows for the assessment of expression shifts and the extraction of biological signals across conditions. The performance of scPCA is evaluated using both simulated and real scRNA-seq datasets, specifically in terms of its ability to model batch effects, extract cell-type-specific factors, and handle multiple time-point data, among other aspects. Below are my comments on the current manuscript version.

Major Comments:

1. Although the model incorporates categorical covariates representing conditions (e.g., drug treatment or genetic perturbation), it currently only considers the condition mean. The distinction between batch effects and true biological signals remains unclear. My concern is how the model differentiates these two factors. Section 2 of the Results demonstrates that scPCA can model batch effects and shows good performance, but the interpretation of these results is not fully explained. How can we confidently distinguish batch effects from biological signals in the model?

We thank the reviewer for this important and thoughtful comment. We have addressed this point in detail in our response to **Reviewer 1**, where we clarify how scPCA distinguishes between batch effects and biological (perturbation) signals, and under which conditions this distinction is valid.

Critically, **scPCA relies on a balanced experimental design**, where all combinations of batch and perturbation conditions are represented in the dataset. This structure enables the model to statistically disentangle variation that is shared across batches (i.e., biological signal) from variation that is specific to individual batches (i.e., technical effects).

In the revised manuscript, we now explicitly discuss these assumptions and illustrate the successful separation of batch and perturbation effects using two representative examples, as described in the Supplementary Methods.

2. For the interferon-beta stimulation dataset in PBMCs, we would expect to see results where one or two cell types, such as CD14-positive monocytes, are clearly associated with IFN-beta stimulation. These results should ideally be visualized in a UMAP plot (e.g., Figures 2b or 2e), where perturbation-associated cell types should show distinct separation. However, some of the perturbation-associated cell types do not appear well-aligned in the UMAP, which needs further clarification.

We thank the reviewer for this thoughtful observation. We agree that in the interferon-beta (IFN- β) stimulation dataset, **CD14⁺ monocytes**, which are known to mount a strong response to IFN- β , are expected to show transcriptional shifts that reflect the perturbation.

In scPCA, perturbation effects are modeled explicitly through a condition-specific transformation encoded in the loading tensor. This formulation allows scPCA to “**factor out**” **the perturbation effect**, aligning cells of the same type—such as CD14⁺ monocytes—across conditions when visualized in the UMAP space. As a result, CD14⁺ monocytes appear well-aligned, even though their transcriptional profiles reflect distinct underlying perturbation dynamics. This alignment indicates successful **integration of shared cell identity** while capturing **perturbation-associated variation in the model parameters**.

We also acknowledge the reviewer’s observation that **megakaryocytes appear less well-integrated**. This is an important point. In scPCA, alignment depends on the assumption that cell types across conditions share sufficient variation in the reference factors. If a given cell type under a perturbation (e.g., megakaryocytes in the stimulated condition) displays a

gene expression profile that is largely orthogonal to the shared reference structure, then scPCA may fail to integrate it.

3. How are the LWDs (Loading Weight Differences) computed in the model? Please provide more detail on this process.

We thank the reviewer for their interest in the computation of Loading Weight Differences (LWDs). We have now added a **more detailed explanation** including a visualisation of this procedure in the **Supplementary Methods** section of the revised manuscript:

“... The Loading Weight Difference (LWD) is defined as the (elementwise) matrix difference between conditions (here the treatment and control directions). In this example, it corresponds exactly to

$$\text{LWD}^{\text{trt}} = \mathbf{W}^{\text{trt}} - \mathbf{W}^{\text{ctl}} = \mathbf{W}^2$$

Conceptually, LWDs quantify the magnitude and direction of the perturbation-induced shift in the latent space, capturing how each factor is altered under a given condition. In practice, the resulting LWD matrix highlights which genes are most affected by a perturbation along specific latent components. These values can be ranked to identify perturbation-responsive genes, and are used for downstream analyses such as gene set enrichment.”

4. In the simulations, the authors should demonstrate scPCA's ability to model batch effects, extract cell-type-specific factors, etc. Additionally, I am concerned about the use of root mean squared error (RMSE) between PW and W in the loss function. Since PW and W come from two different spaces, they may represent different variables, which raises the question of how appropriate it is to compute RMSE between them. These variables may differ substantially but still be reasonable in the context of their respective spaces.

We thank the reviewer for this insightful comment and the opportunity to clarify.

Regarding the first point: we have added **simulation experiments** in response to **Reviewer 1**, specifically designed to demonstrate that scPCA can model batch effects under a balanced experimental design. These simulations show how the model separates technical variation (e.g., shifts or rotations across batches) from true biological signals.

In addition, the real-data examples presented throughout the manuscript demonstrate scPCA's ability to extract **meaningful, cell-type-specific factors**. For instance, in the analysis of lung cells from aging mice and brain cells from light-stimulated animals, scPCA identifies cell type-specific transcriptional responses — highlighting its utility in capturing structured biological variation across heterogeneous cell-type populations.

Second, we would like to clarify a potential misunderstanding: **the RMSE is not part of the scPCA loss function**. The model is trained by **maximizing the Evidence Lower Bound (ELBO)**, which is based on a **Negative Binomial likelihood** (see also the answer to Reviewer 1) of the observed gene expression counts. The ELBO governs all learning and optimization steps.

5. The authors should include a comparison with CellANOVA (Zhaojun Zhang et al., 2024, Nature Biotechnology) to provide a broader context for the performance of scPCA.

We thank the reviewer for this helpful suggestion. CellANOVA (Zhang et al., 2024, Nature Biotechnology) is a valuable recent method for modeling condition-specific variation in single-cell data, and we agree that comparing it to scPCA provides useful context for understanding our model's unique contributions. We now cite and discuss CellANOVA explicitly in the revised manuscript (**Supplementary Methods**):

“CellANOVA²³ is a post-hoc, projection-based method that builds on precomputed low-dimensional embeddings—typically derived from tools like Harmony²⁴ or Seurat²⁵. It applies an ANOVA-style residual decomposition to partition variation into shared, batch-associated, and condition-specific factorisations.

By contrast, scPCA learns a shared low-dimensional basis and models condition effects as explicit linear transformations of this reference basis. This enables scPCA to provide interpretable, factor-specific perturbation readouts. Unlike CellANOVA, which attributes variation to residuals without an explicit link to latent factors, scPCA offers a mapping between latent axes and condition-induced changes. As a result, the two frameworks characterize condition effects through fundamentally different representational strategies.”

We note because of these conceptual and technical differences, a direct comparison between CellANOVA and scPCA is not straightforward. We therefore view the two methods as complementary, addressing related but distinct goals in the analysis of multi-condition single-cell datasets. We thank the reviewer for prompting this important clarification.

6. How should one determine the optimal number of scPCs for analysis in real datasets? The methodology for selecting the number of principal components is not sufficiently addressed.

We thank the reviewer for raising this important point. We have addressed this question in our response to **Reviewer 2**, where we detail the procedure used to select the optimal number of scPCs (latent factors). Specifically, we assess model fit using the root mean squared error (RMSE) and observe where this metric saturates, indicating that additional components no longer improve reconstruction. We also report that results are robust to moderate changes in the number of components, and provide further guidance in the revised **Methods** section.

Reviewers' comments:

Reviewer #1 (Remarks to the Author):

I sincerely thank the authors for making this comprehensive revision. However, I still have some concerns about result justification and necessary benchmarking analysis.

1. The authors did not compare scPCA with other baselines for batch effect correction in the quantitative approach. Only one score like iLISI is not a strong proof and it seems that Seurat also did well in all of the four datasets. I would encourage the authors take a look at scIB and include other metrics to reflect the evaluation from both batch and bio perspective.

We thank the reviewer for this thoughtful comment. In line with the suggestion, we applied the **scIB** framework, which integrates several complementary metrics to assess **biological conservation** (Isolated labels, KMeans NMI, KMeans ARI, cLISI) and **batch correction** (Silhouette batch, KBET, Graph connectivity, PCR comparison). We have updated **Supplementary Figure 7c** to display the mean scores for biological conservation, batch correction, and overall performance, offering a more comprehensive assessment of **scPCA** in the simulation experiments.

Supplementary Figure 7: Benchmarking scPCA against PCA, Harmony, Seurat, and scVI on simulated datasets of varying complexity and size. a and b. UMAP visualizations of low-dimensional embeddings obtained using five integration methods (PCA, Harmony, Seurat (CCA), scVI, and scPCA) across two simulation scenarios with increasing dataset sizes (100, 1,000, 10,000, and 100,000 cells). Dots represent cells, colors either batch (blue/orange) or cell type (green/red). **a.** Single-cell simulations (**Methods**) with one cell type and **b.** Simulations involving two distinct cell types. **c.** Average biological conservation (Isolated labels, KMeans NMI, KMeans ARI, cLISI), batch correction (Silhouette batch, KBET, Graph connectivity, PCR comparison) and overall performance scores across the simulation experiments. **d.** Runtime (top row) and memory usage (bottom row) for each method as a function of dataset size, evaluated on both simple and complex simulation scenarios.

Also, could the authors explain why methods such as Harmony show imbalanced performances across other cases, since Harmony is a top-performer in various benchmarking analysis, does this mean Harmony works for real data? I think it is necessary to include one or two real data to justify their contributions.

We thank the reviewer for raising this important point. We emphasize that the results from the simulated benchmarks do **not** imply that Harmony performs worse on **real data**. The poor performance arises because the data were simulated according to the **generative process of the scPCA model** (see Supplementary Methods). Note that in practice **the true data-generating process of real multi-condition single-cell datasets is unknown and unlikely to exactly match the assumptions of any single model** (as George Box famously noted: “All models are wrong, but some are useful”). However, since scPCA explicitly infers parameters under this framework, the simulation setting naturally favors scPCA over alternative methods. Therefore, these simulations are therefore best understood as a validation exercise, demonstrating that scPCA can recover latent factors that perform well with respect to both batch and biological conservation metrics.

To address the reviewer’s suggestion, we have additionally included **benchmarking results** comparing **scPCA’s performance** in terms of **biological conservation and batch correction** against other tools, including **Harmony, CellANOVA, Seurat, and scVI**, on **real datasets**. The results of these analyses are now presented in **Extended Data Figures 7 and 8**, providing a comprehensive view of scPCA’s performance relative to established methods in practical and controlled settings.

Extended Data Fig. 7: Comparison of scPCA embeddings with other integration methods on published single-cell datasets. a. and b UMAP visualizations of cell embeddings generated by PCA, Harmony (harmony), CellANOVA (cnova), Seurat (seurat), and scVI (scvi) across datasets from Kang et al. (kang), Zheng et al. (zheng), Hrvatin et al. (hrvatin), and Wagner et al. (wagner) colored by condition variable (a) and cell types (b).

a Kang et al.

Method	Bio conservation					Batch correction					Aggregate score		
	Isolated labels	KMeans NMI	KMeans ARI	Silhouette label	cLISI	Silhouette batch	iLISI	KBET	Graph connectivity comparison	PCR	Batch correction	Bio conservation	Total
X_seurat	1.00	1.00	1.00	1.00	1.00	0.89	0.42	0.96	0.98	0.99	0.85	1.00	0.94
X_scPCA	0.75	0.91	0.74	0.55	1.00	0.97	0.93	1.00	0.95	1.00	0.97	0.79	0.86
X_harmony	0.64	0.88	0.68	0.73	1.00	0.92	0.78	0.93	0.90	0.98	0.90	0.78	0.83
X_scvi	0.49	0.64	0.44	0.36	1.00	1.00	0.40	0.50	1.00	0.75	0.73	0.58	0.64
X_pca	0.51	0.69	0.42	0.47	1.00	0.00	0.00	0.00	0.91	0.00	0.18	0.62	0.44
X_cnova	0.00	0.00	0.00	0.00	0.00	0.87	1.00	0.77	0.00	0.96	0.72	0.00	0.29

b Zheng et al.

Method	Bio conservation					Batch correction					Aggregate score		
	Isolated labels	KMeans NMI	KMeans ARI	Silhouette label	cLISI	Silhouette batch	iLISI	KBET	Graph connectivity comparison	PCR	Batch correction	Bio conservation	Total
X_harmony	1.00	1.00	1.00	1.00	1.00	0.92	0.71	1.00	0.80	0.00	0.69	1.00	0.87
X_scPCA	0.45	0.96	0.98	0.45	1.00	0.94	0.64	0.71	0.00	0.51	0.56	0.77	0.68
X_scvi	0.17	0.06	0.07	0.17	1.00	1.00	0.64	0.63	0.77	0.83	0.77	0.30	0.49
X_pca	0.96	0.01	0.00	0.96	1.00	0.00	0.00	0.00	1.00	0.00	0.20	0.59	0.43
X_seurat	0.00	0.00	0.00	0.00	0.00	0.44	1.00	0.47	0.05	1.00	0.59	0.00	0.24

c Angelidis et al.

Method	Bio conservation					Batch correction					Aggregate score		
	Isolated labels	KMeans NMI	KMeans ARI	Silhouette label	cLISI	Silhouette batch	iLISI	KBET	Graph connectivity comparison	PCR	Batch correction	Bio conservation	Total
X_seurat	0.76	1.00	1.00	1.00	1.00	0.43	1.00	0.50	0.47	1.00	0.68	0.95	0.84
X_scPCA	0.77	0.83	0.38	0.64	0.90	0.72	0.00	0.00	0.88	0.00	0.32	0.70	0.55
X_cnova	1.00	0.27	0.61	0.42	0.27	0.00	0.89	1.00	0.00	0.97	0.57	0.52	0.54
X_scvi	0.52	0.06	0.08	0.00	0.39	1.00	0.66	0.38	1.00	0.54	0.72	0.21	0.41
X_harmony	0.00	0.00	0.00	0.35	0.00	1.00	0.70	0.86	0.36	0.72	0.73	0.07	0.33
X_pca	0.01	0.07	0.15	0.34	0.15	0.66	0.15	0.11	0.47	0.00	0.28	0.15	0.20

d Hrvatin et al.

Method	Bio conservation					Batch correction					Aggregate score		
	Isolated labels	KMeans NMI	KMeans ARI	Silhouette label	cLISI	Silhouette batch	iLISI	KBET	Graph connectivity comparison	PCR	Batch correction	Bio conservation	Total
X_harmony	0.48	0.96	0.96	0.92	1.00	1.00	0.96	1.00	0.93	0.85	0.95	0.86	0.90
X_seurat	1.00	1.00	1.00	1.00	1.00	0.35	0.59	0.27	0.85	0.86	0.59	1.00	0.83
X_pca	0.47	0.85	0.72	0.91	1.00	0.81	0.83	0.70	0.95	0.00	0.66	0.79	0.74
X_scPCA	0.29	0.88	0.80	0.35	1.00	0.75	0.00	0.00	0.86	1.00	0.52	0.67	0.61
X_scvi	0.00	0.63	0.39	0.00	1.00	0.83	0.71	0.56	1.00	0.00	0.62	0.41	0.49
X_cnova	0.55	0.00	0.00	0.29	0.00	0.00	1.00	0.83	0.00	0.00	0.37	0.17	0.25

e Wagner et al.

Method	Bio conservation					Batch correction					Aggregate score		
	Isolated labels	KMeans NMI	KMeans ARI	Silhouette label	cLISI	Silhouette batch	iLISI	KBET	Graph connectivity comparison	PCR	Batch correction	Bio conservation	Total
X_scvi	0.89	0.84	0.74	0.97	1.00	0.94	0.31	0.89	1.00	0.52	0.73	0.89	0.82
X_scPCA	0.98	0.91	0.80	1.00	1.00	0.72	0.00	0.77	0.97	0.32	0.56	0.94	0.79
X_harmony	0.62	0.91	0.77	0.76	0.97	0.74	0.52	1.00	0.97	0.38	0.72	0.81	0.77
X_seurat	1.00	1.00	1.00	0.50	1.00	0.00	0.38	0.91	0.95	0.57	0.56	0.96	0.76
X_pca	0.61	0.91	0.83	0.75	0.97	0.61	0.31	0.74	0.98	0.00	0.53	0.81	0.70
X_cnova	0.00	0.00	0.00	0.00	0.00	1.00	1.00	0.00	0.00	1.00	0.60	0.00	0.24

Extended Data Fig. 8: Benchmarking of scPCA and alternative integration methods on published single-cell datasets using scIB metrics. a-e. Comparison of biological conservation and batch correction performance across published datasets from Kang et al. (a), Zheng et al. (b), Angelidis et al. (c), Hrvatin et al. (d), and Wagner et al. (e). Methods compared include scPCA (X_{scpca}), Seurat (X_{seurat}), Harmony ($X_{harmony}$), scVI (X_{scvi}), PCA (X_{pca}), and CellANOVA (X_{cnova}).

To reflect the inclusion of these additional benchmarking analyses, we have amended the **Results** section of the manuscript as follows:

*“... We further benchmarked scPCA against other single-cell latent variable models using both simulation experiments (**Supplementary Fig. 7a-c**) and a diverse set of published datasets (**Extended Data Figure 7 and 8**). Across these analyses, scPCA consistently demonstrated competitive performance across a comprehensive range of evaluation metrics.”*

2. I understand CINEMA-OT cannot be included in this research, but I did not find reasons for not including CellANOVA here. I do not believe that CellANOVA will be much affected by the results from Harmony as it is designed to tackle the missing covariate problems caused by Harmony. I think it is necessary to include CellANOVA in the benchmarking analysis.

We thank the reviewer for this valuable comment and the suggestion to include **CellANOVA** in our benchmarking analysis.

We would like to clarify that **CellANOVA** relies on **Harmony** (<https://www.nature.com/articles/s41587-024-02463-1#Sec17>) to compute cell embeddings and subsequently performs **post hoc regression** to disentangle biological and technical sources of variation in complex scRNA-seq experiments. As such, **CINEMA-OT** inherently depends on Harmony’s integration step and cannot be applied independently to datasets where Harmony embeddings are not available. We note that **Harmony** itself is included in our benchmarking analysis.

However, we observed that the authors of **CellANOVA** perform **principal component analysis (PCA)** on the “denoised” single-cell expression matrix obtained after their adjustment step (see https://github.com/Janezjz/cellanova/blob/main/tutorials/cellanova_integration.ipynb, Section 7. UMAP Visualization). In our revised benchmarks, we have now incorporated **PCA factors derived from the CellANOVA output**, and computed **UMAP** as well as **scIB metrics** to evaluate the **biological conservation** and **batch correction** of these embeddings. The results of these experiments are shown in **Extended Data Figure 7 and 8**. We note that **CellANOVA** was not applicable to the **Zheng** dataset, as this dataset lacks annotations for technical replicates. Likewise, **CellANOVA** could not be applied to our **simulation experiments** shown in **Supplementary Figure 7**, since **scPCA’s generative model** does not explicitly model technical replicates.

Reviewer #2 (Remarks to the Author):

The authors have addressed all of my major concerns. Before publication, I have one minor suggestion:

It is commendable that the authors included a fair discussion on the limitations of scPCA in capturing non-linear changes. Supplementary Figure 7 is particularly important in demonstrating the performance of the proposed method, and I recommend that it be included in the main manuscript.

We sincerely thank the reviewer for the positive feedback and the thoughtful suggestion. We appreciate the recognition of our discussion on the limitations of **scPCA** in capturing non-linear effects and agree that this is an important aspect of the method. While we chose to keep **Supplementary Figure 7** in the supplementary materials to preserve the overall structure and flow of the manuscript, we note that this limitation and its implications are **discussed and emphasized** in the **Discussion section**.

Reviewer #3 (Remarks to the Author):

No further comments on it.

We sincerely thank the reviewer for taking the time to evaluate our manuscript and for their positive assessment. We are pleased that no further comments were raised.

REVIEWERS' COMMENTS:

Reviewer #1 (Remarks to the Author):

Thank you for your great work. I will support the publication of this paper.

We thank the reviewer for their positive assessment of our work and for their support of its publication. We greatly appreciate the encouraging feedback.